# Intraneural stimulation elicits discrimination of textural features by artificial fingertip in intact and amputee humans

Calogero Maria Oddo[1*†], Stanisa Raspopovic[1,2,3†], Fiorenzo Artoni[1,2], Alberto Mazzoni[1], Giacomo Spigler[1], Francesco Petrini[2,3,4,5], Federica Giambattistelli[6], Fabrizio Vecchio[5], Francesca Miraglia[5], Loredana Zollo[4], Giovanni Di Pino[4,6], Domenico Camboni[1], Maria Chiara Carrozza[1], Eugenio Guglielmelli[4], Paolo Maria Rossini[5,7], Ugo Faraguna[8,9,10], Silvestro Micera[1,3,2*]

[1]The BioRobotics Institute, Scuola Superiore Sant'Anna, Pisa, Italy; [2]Bertarelli Foundation Chair in Translational NeuroEngineering, Institute of Bioengineering, School of Engineering, École Polytechnique Fédérale de Lausanne, Lausanne, Switzerland; [3]Center for Neuroprosthetics, École Polytechnique Fédérale de Lausanne, Lausanne, Switzerland; [4]Laboratory of Biomedical Robotics & Biomicrosystems, Università Campus Bio-Medico di Roma, Roma, Italy; [5]Brain Connectivity Laboratory, IRCCS San Raffaele Pisana, Roma, Italy; [6]Institute of Neurology, Università Campus Bio-Medico di Roma, Roma, Italy; [7]Institute of Neurology, Catholic University of The Sacred Heart, Roma, Italy; [8]Azienda Ospedaliero-Universitaria Pisana, Pisa, Italy; [9]IRCCS Stella Maris Foundation, Pisa, Italy; [10]Dipartimento di Ricerca Traslazionale e delle Nuove Tecnologie in Medicina e Chirurgia, Università di Pisa, Pisa, Italy

*For correspondence: calogero. oddo@sssup.it (CMO); silvestro. micera@epfl.ch;silvestro.micera@ sssup.it (SM)

†These authors contributed equally to this work

**Abstract** Restoration of touch after hand amputation is a desirable feature of ideal prostheses. Here, we show that texture discrimination can be artificially provided in human subjects by implementing a neuromorphic real-time mechano-neuro-transduction (MNT), which emulates to some extent the firing dynamics of SA1 cutaneous afferents. The MNT process was used to modulate the temporal pattern of electrical spikes delivered to the human median nerve via percutaneous microstimulation in four intact subjects and via implanted intrafascicular stimulation in one transradial amputee. Both approaches allowed the subjects to reliably discriminate spatial coarseness of surfaces as confirmed also by a hybrid neural model of the median nerve. Moreover, MNT-evoked EEG activity showed physiologically plausible responses that were superimposable in time and topography to the ones elicited by a natural mechanical tactile stimulation. These findings can open up novel opportunities for sensory restoration in the next generation of neuro-prosthetic hands.

## Introduction

One of the most remarkable characteristics of a human hand is its ability to gather a rich variety of sensory information about the external world. In particular, tactile information conveyed by the four classes of low-threshold mechanoreceptor afferent units in the fingertips (*Johnson et al., 2000*;

**eLife digest** Our hands provide us with a wide variety of information about our surroundings, enabling us to detect pain, temperature and pressure. Our sense of touch also allows us to interact with objects by feeling their texture and solidity. However, completely reproducing a sense of touch in artificial or prosthetic hands has proven challenging. While commercial prostheses can mimic the range of movements of natural limbs, even the latest experimental prostheses have only a limited ability to 'feel' the objects being manipulated. Oddo, Raspopovic et al. have now brought this ability a step closer by exploiting an artificial fingertip and appropriate neural interfaces through which different textures can be identified.

The initial experiments were performed in four healthy volunteers with intact limbs. Oddo, Raspopovic et al. connected the artificial fingertip to the volunteers via an electrode inserted into a nerve in the arm. When moved over a rough surface, sensors in the fingertip produced patterns of electrical pulses that stimulated the nerve, causing the volunteers to feel like they were touching the surface. The volunteers were even able to tell the difference between the different surface textures the artificial fingertip moved across.

The temporary electrodes used in this group of volunteers are unsuitable for use with prosthetic limbs because they can easily be knocked out of position. Therefore, in a further experiment involving a volunteer who had undergone an arm amputation a number of years previously, Oddo, Raspopovic et al. tested an implanted electrode array that could, in principle, remain in place long-term. This volunteer could also identify the different textures the artificial fingertip touched, with a slightly higher degree of accuracy than the previous group of intact volunteers. Further studies are now required to explore the potential of this approach in larger groups of volunteers.

*Johnson, 2001*; *Vallbo and Johansson, 1984a*; *Abraira and Ginty, 2013*) is fundamental for manipulation activities (*Edin et al., 1992*; *Edin and Johansson, 1995*). In addition, the biological sensors innervating the glabrous skin can provide complex types of information such as the onset of the contact with an object, the level of performed grasping force, and discrimination of textural features (*Weber et al., 2013*; *Darian-Smith et al., 1980a*).

The possibility of providing natural and rich sensory information to hand prosthesis users represents a major achievement. In fact, even though a significant research in biology, medicine and engineering has produced artificial hands that progressively approached the performance of a "natural hand", it is a shared belief that providing a sense of touch is still the missing milestone (*Kwok, 2013*). This achievement will allow the "symbiosis" between the user and interface to become more adaptive (to changing tasks and situations), more robust (beyond interfering stimuli), more effective (learn from the past to anticipate or predict the future), and more natural (rapidly becoming part of the body scheme). Therefore, the restoration of sensory perception is the crucial step to achieve in the development of next generation of artificial limbs and hand prostheses, in particular.

Thus far, promising results have been recently achieved to restore information about the touch of objects (*Tan et al., 2014*) and also the level of produced grasping force (*Raspopovic, 2014*). The restoration of ability to judge textural features represents the next significant step towards the re-establishment of close-to-natural sensory skills of a natural hand.

Here, we sought to achieve this goal via an integrated approach to mimic natural coding using a neuromorphic, real-time, mechano-neuro-transduction (*Spigler et al., 2012*) (MNT) process through a sensorized artificial finger (*Oddo et al., 2011a*) that integrates a Micro Electro-Mechanical System (MEMS) tactile sensor (*Beccai et al., 2005*). In this framework, the temporal coding of tactile information was based on the use of a biologically plausible neural model (*Izhikevich, 2003*) , which has shown promise as a versatile and computationally efficient framework for reproducing a wide range of phenomenological neural responses to stimuli.

In this study, the MNT process was first tested in intact subjects by delivering electrical stimulation to their sensory peripheral nerve fibers during microstimulation via tungsten needle microelectrodes (*Vallbo et al., 1984b*; *Torebjörk et al., 1987*). Furthermore, the effects of natural mechanical

stimulation at the fingertip and of the MNT-based electrical stimulation were qualitatively compared in terms of electrophysiological signals elicited in the contralateral sensory cortex of the subjects.

Moreover, we wanted to evaluate whether the results achieved during needle microstimulation could be translated into experiments, which would be related to the real-time and longer-term use of hand prostheses. In fact, percutaneous needle microstimulation of peripheral nerves cannot be used in amputees as part of an effective long-term assistive device because a stable needle-to-fibers spatial relationship cannot be maintained in a moving limb. Therefore, its use can only be limited to a single experimental session lasting a few hours at most (*Vallbo et al., 2004*). Instead, implantable neural interfaces, such as Transverse Intrafascicular Multichannel Electrodes (TIMEs) (*Boretius et al., 2010*) , are surgically positioned and firmly stabilized to adhere to nerve fascicles and do not require an arm rest compared with needle interfaces. Therefore, implantable neural interfaces are suitable for long-term implants. In this study, we developed a novel computational model to investigate the similarities between neural effects, such as in afferent nerve recruitment curves, achieved using microstimulation with percutaneous needles and with TIME neural interfaces. This comparison allowed verification of the possibility of extending the results achieved using percutaneous needle stimulation to TIMEs.

The MNT process was tested in acute conditions in four intact subjects and was validated in a TIMEs implanted subject with a transradial amputation. For the first time in human hand neuroprosthetics, these integrated approaches allowed to show that discrimination of textural features can be reliably provided to users in different experimental conditions using peripheral intraneural electrical stimulation. The range of tested tactile stimuli in the current work is on the order of millimeters (0.5–3.0 mm). Thus, these stimuli pertain to the lower boundary (towards the fine region) of stimuli that are typically classified as coarse (*Weber et al., 2013*).

## Results

### Experiments with intact subjects using needle microstimulation of the median nerve

The MNT process translates surface coarseness into the injection of current pulses into the nerve. It qualitatively mimics the neuronal activity recorded during human microneurographic experiments (*Oddo et al., 2011b*). The MNT approach was initially tested in four intact volunteers using percutaneous electrical microstimulation of the median nerve (*Vallbo et al., 1984b*; *Torebjörk and Ochoa, 1980*) (*Figure 1a*, *Figure 2*). The participants - without visual or acoustic cues about the stimuli - were asked to discriminate surface pairs (*Figure 1b*) that differed in the Spatial Period (SP) of alternating ridges and grooves (gratings), i.e., in the distance between consecutive ridges separated by grooves (defined in *Figure 2a*), which was a constant quantity in each half grating (as shown in *Figure 1b*).

Via percutaneous electrical neural microstimulation, they reported mechanical sensation pertaining to the palmar side of the first four fingers of the hand. Microstimulation allowed users to reach discrimination ability above 77% (107/138, *Figure 1c*, *Figure 3a*) during a three-alternative forced-choice (3AFC) psychophysical procedure (*Perez et al., 2010*; *Gibson and Craig, 2005*) mediated by the artificial touch system, which is based on the use of a MEMS sensor embedded into a human-sized robotic fingertip (*Video 1*). Confidence analyses indicated that percutaneous electrical microstimulation successfully induced percepts that were used to assess the coarseness of textured surfaces (*Figure 3b*). The capability to discriminate between the two sides of the surface pairs was correlated with the difference between their spatial periods (*Figure 3c*).

As described hereafter, the comparison between the EEG activity that was evoked by the natural mechanical tactile stimulation of the real fingertip in the right hand and the one evoked by the substitutive electrical stimulation showed no significant differences in source topography, response timing, and clustering of cortical connections between the two stimulation modalities. Event-related potentials (*Figure 4a*) after substitutive electrical (n = 4, estimated power 0.75, *Figure 4—figure supplement 2*) and natural mechanical stimulation (n = 4, estimated power 0.79, *Figure 4—figure supplement 3*) conditions did not reveal any statistical difference (Montecarlo statistics with cluster correction for multiple comparisons). Furthermore, a network graph analysis approach (*Vecchio et al., 2015a*) revealed a lateralized EEG frequency modulation that was evoked both by

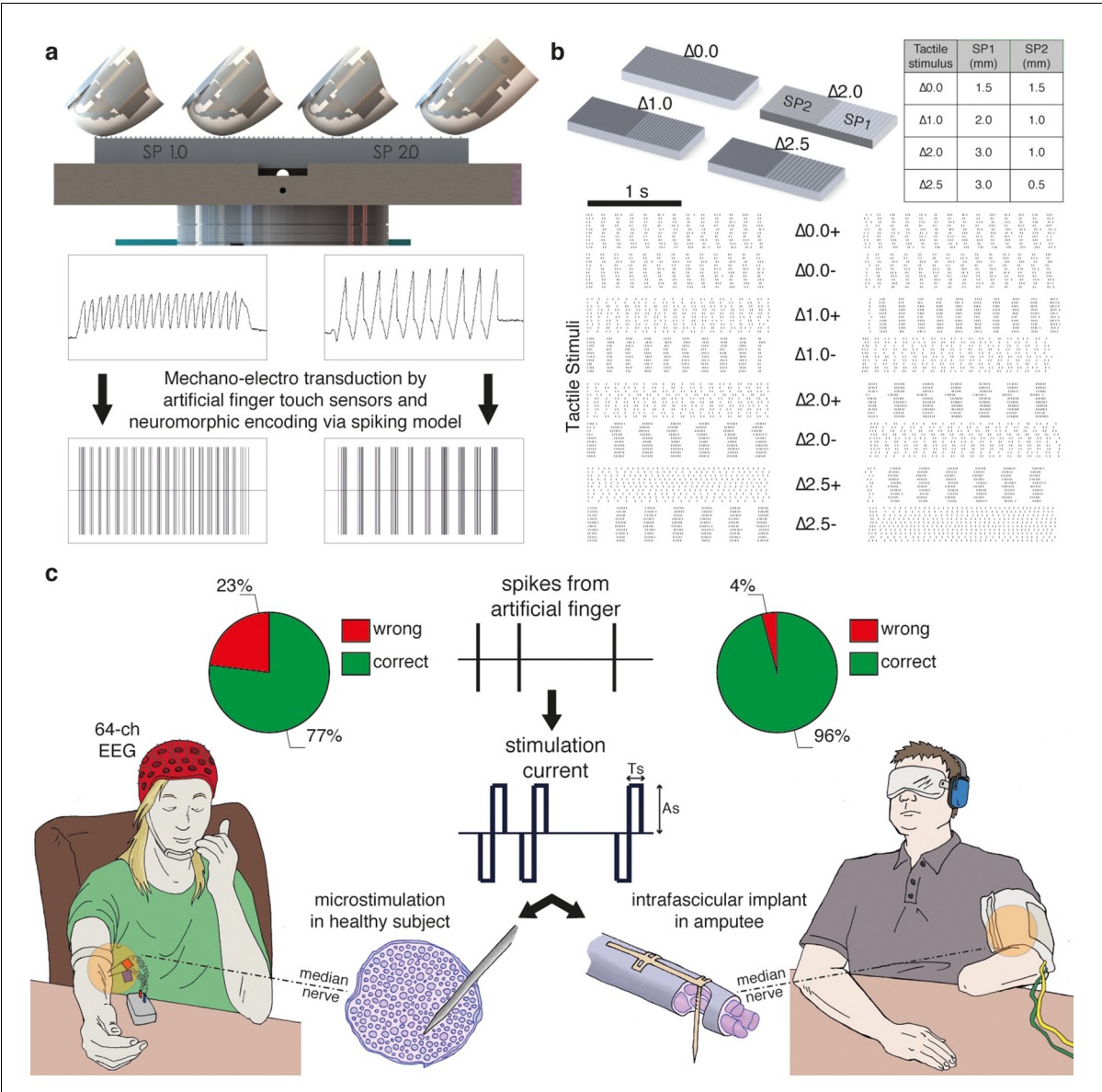

**Figure 1.** Experimental setup and performance metrics. (a) Sensorized artificial finger and tactile stimulation platform. (b) Tactile stimuli that were used in the three-alternative forced-choice (3AFC) psychophysical protocol and the raster plot of spike trains that were generated in all sessions with one subject by the artificial finger while the gratings were slid. (c) Setup of percutaneous electrical microstimulation (left) and implanted intrafascicular stimulation (right) of the median nerve, and discrimination performance during all experimental sessions involving four intact subjects and one transradial upper limb amputee. Source data of the spike trains that were transduced by the artificial finger while the gratings were indented and slid over have been deposited in Dryad (*Oddo et al., 2016*). Such spikes were used to trigger the neural stimulator in all the experimental sessions with DAS amputee (raster plot depicted in *Figure 1b*).

electrical and mechanical stimuli (*Figure 4b*). Indeed, the primary sensorimotor areas in the hemisphere contralateral to the stimulus presented a significant reduction (3-way ANOVA followed by Duncan's multiple range test, $F(1,6) = 6.48$, $p<0.05$, comparison to the ipsilateral hemisphere) in the clustering coefficient following the incoming sensory stimulus, regardless of its tactile or substitutive nature (*Figure 4c*). Moreover, the generator sources of short-latency components of Somatosensory Evoked Potentials (SEPs) that were elicited by the substitutive electrical stimulation were localized at the Postcentral Gyrus (Brodmann Areas 2 and 3), which was consistent with a physiological tactile

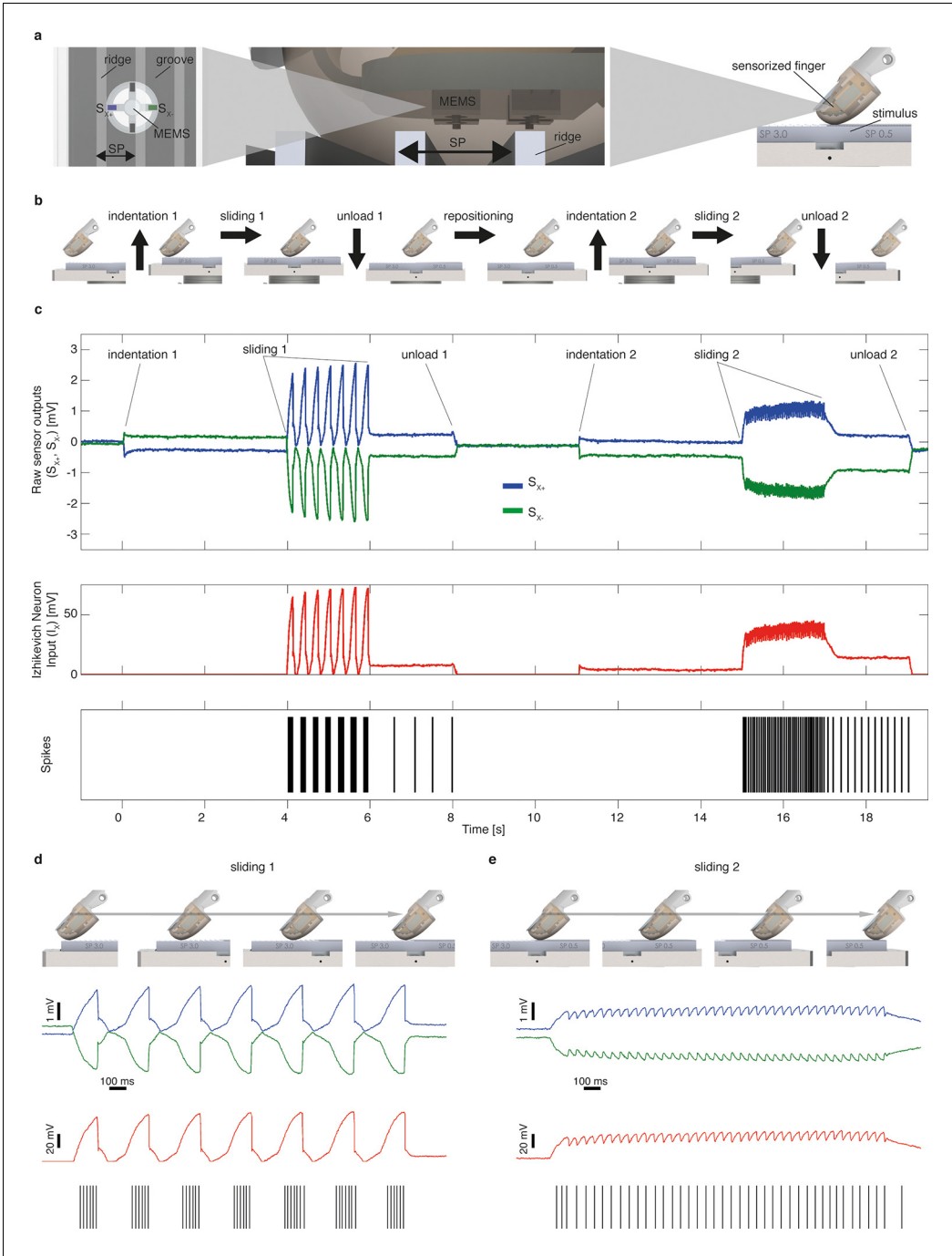

**Figure 2.** Mechano-neuro-transduction process. (**a**) MEMS sensor with 4 transducing piezoresistors implanted at the base of a cross-shaped structure (sensor piezoresistive outputs $S_{x+}$ and $S_{x-}$ are represented in blue and in green, respectively), grating with ridges and grooves that alternate with spatial period SP, and sensorized fingertip, which is in contact with tactile stimulus. (**b**) The sequence of presentation of surface pair to the artificial finger. (**c**) Example of implementation of the Izhikevich model for real-time conversion of MEMS sensor data into a sequence of artificial neural spikes. The blue and green traces show raw sensor outputs from the pair of opposing channels depicted in panel a. The red trace shows the input to the Izhikevich artificial neuron, which results from the application of *Equation 1 and 2* (described in the Materials and methods section). The black lines depict the spikes that are generated when the membrane potential of the Izhikevich artificial neuron reaches the threshold (*Equation 5* in the Materials and methods section) and, thus, the neural stimulator is triggered. (**d–e**) Implementation of the Izhikevich model with a close-up view during the sliding motion over the first and second halves of the grating. The traces and color-coding are shown in panel c.

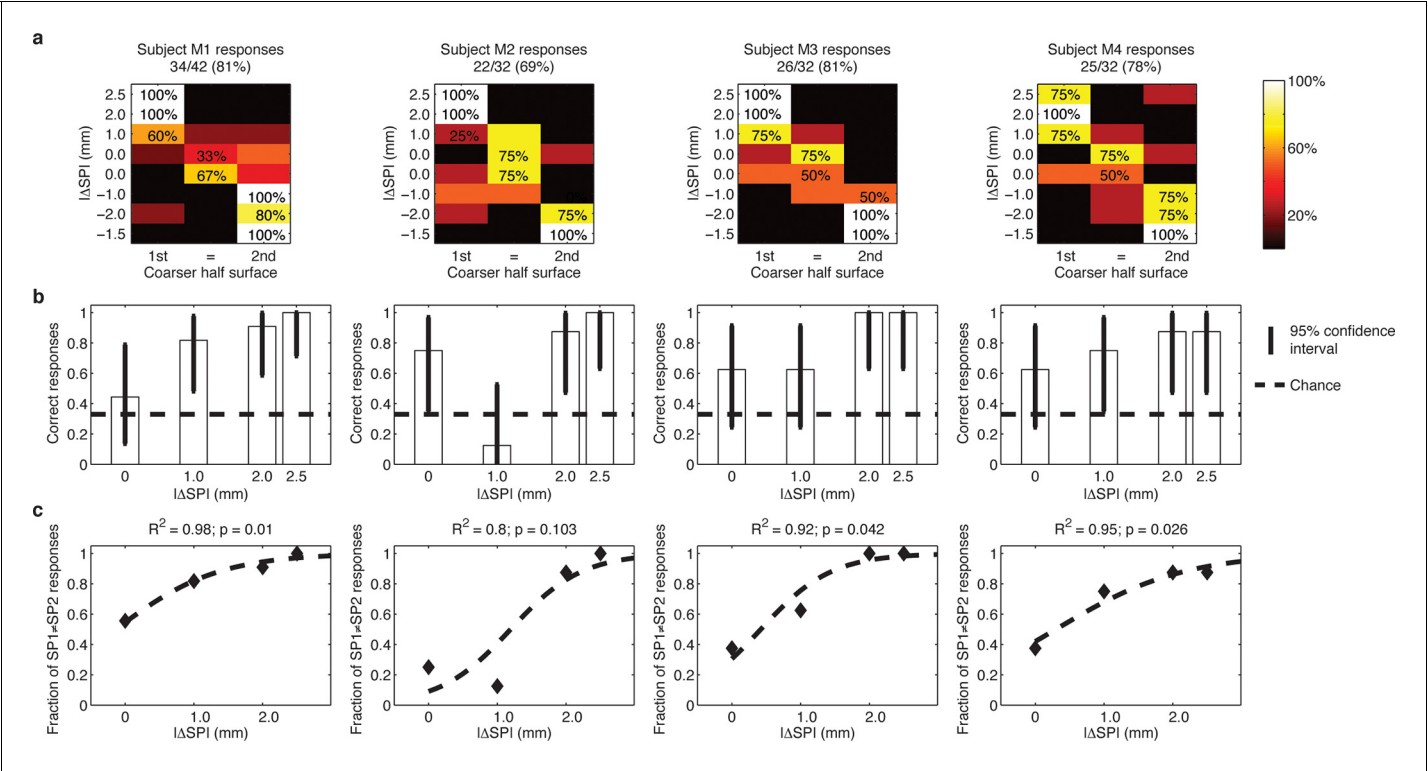

**Figure 3.** Responses of intact subjects during the 3AFC psychophysical protocol with percutaneous electrical microstimulation of the median nerve. Each column reports results of the analyses on individual subject basis. (a) Each panel displays the confusion matrix of behavioral responses relative to the four intact subjects with microstimulation. The titles indicate correct/total responses (percentage). (b) Vertical bars display correct responses that are associated with each stimulus. The vertical solid lines over each bar indicate the 95% confidence intervals (Clopper Pearson exact interval) per stimulus. The dashed horizontal line indicates chance level (1/3). (c) The fraction of trials a pair of stimuli is perceived as different as a function of difference in spatial period (ΔSP) between the two stimulus halves, and logistic fit (dashed line). The title reports the $R^2$ associated with the fit, i.e., the fraction of data variance explained by the logistic function, and the significance of the Pearson correlation between data points and the fit.

activation of the primary somatosensory cortex (*Figure 5*), as previously described using a hand area functional source separation method (*Di Pino et al., 2012*).

## Translatability from needle microstimulation to TIME-based stimulation

The possibility to translate results from needle microstimulation to TIMEs was investigated by developing a novel hybrid model (*Raspopovic et al., 2011*) of the median nerve (*Figure 6*) with both a microneedle and TIME interfaces inserted inside the nerve trunk (*Figure 7*, *Figure 8*). The model, which takes into account realistic anatomical (*Jabaley et al., 1980*) and neurophysiological (*Vallbo et al., 1984*) data, indicated that the stimulated portion of axonal population (*Figure 7*), and, therefore, the type of sensation, together with the stimulation threshold (statistically non-different: $p > 0.05$, Kruskal-Wallis test) were similar for the two interfaces. The range of electric charge necessary for recruitment in both cases was comparable (*Figure 7*, *Figure 8*). These findings provided evidence that implanted intraneural electrodes could achieve results that were comparable to needle

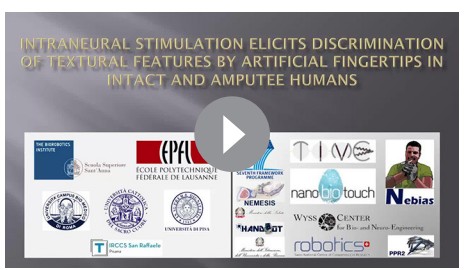

**Video 1.** An example of the 3AFC psychophysical experiment with implanted intrafascicular stimulation of DAS amputee (as illustrated in *Figure 1*). The video includes an interview with DAS amputee subject reporting the percepts immediately after one experimental session.

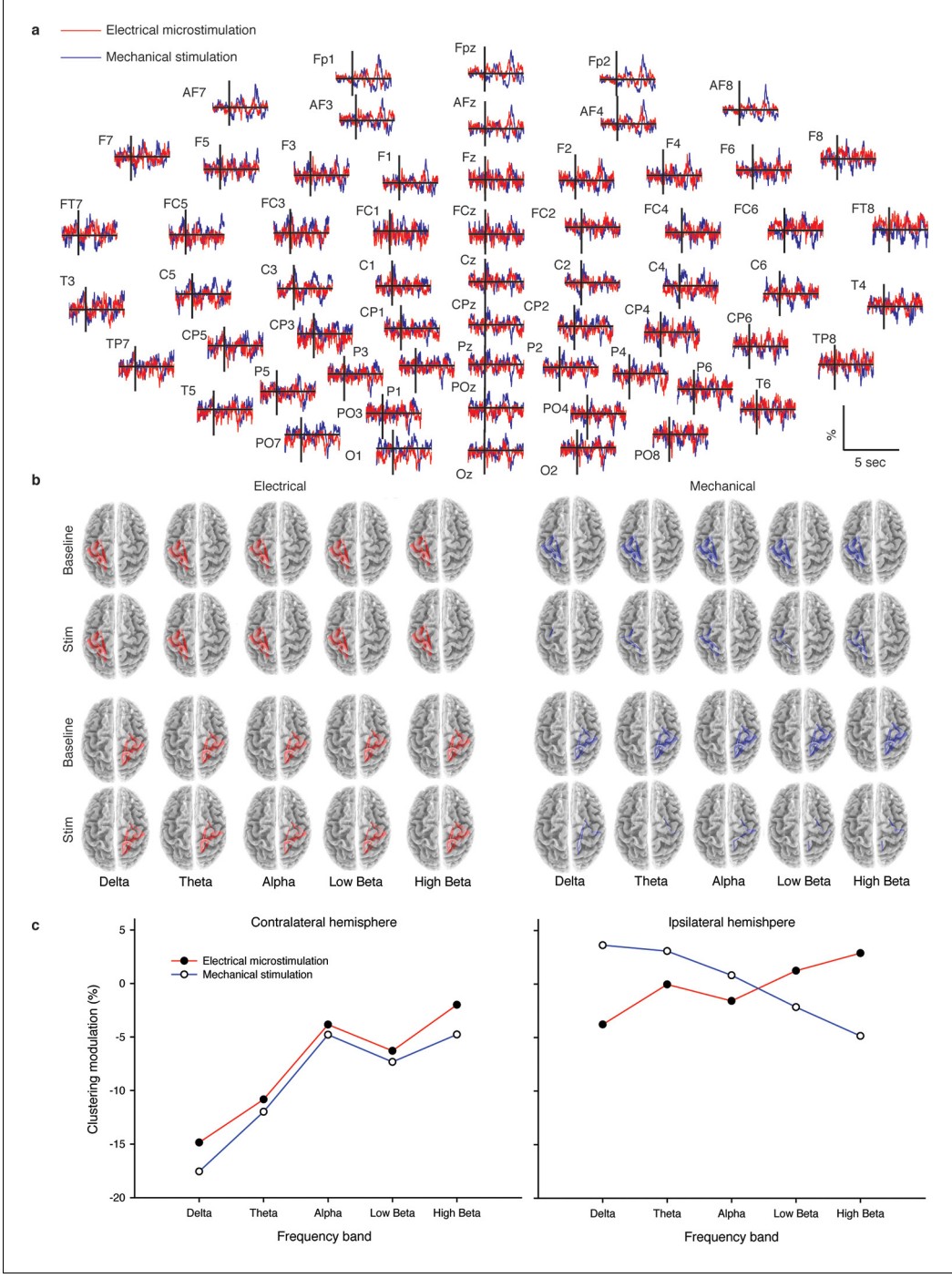

**Figure 4.** Cortical response to mechanical and electrical stimulation using a surface with 1.5 mm SP. (**a**) Grand average event related potentials (ERPs) of all subjects (n = 4) for both substitutive neuromorphic electrical (red) and natural mechanical tactile (blue) stimulation, ranging from -1500 to 3000 ms with respect to the stimulus onset. Each channel was normalized for the standard deviation of the prestimulus. (**b**) eLORETA connectivity maps for delta, theta, alpha, low beta and high beta bands. Each tract (red for electrical and blue for mechanical stimulations) among the 7 sensorimotor regions of interest (Brodmann Areas BA 1–7) reports the connectivity value higher than the cut-off threshold (functional coupling >0.3). (**c**) Clustering modulation (percentage of variation during stimulation with respect to baseline) in the left and right hemispheres with electrical and mechanical stimulations. A significant reduction in clustering modulation across all frequencies occurred in the hemisphere contralateral to the stimulation (p<0.05, comparison to the right hemisphere, Duncan test after ANOVA).

*Figure 4 continued on next page*

*Figure 4 continued*

The following figure supplements are available for figure 4:

**Figure supplement 1.** Grand average event related potentials (ERPs) of all subjects (n = 4) at the FC1 electrode for both the substitutive neuromorphic electrical (red) and natural mechanical tactile (black) stimulation, in the -150 to 350 ms window with respect to stimulus onset, with confidence interval bars.

**Figure supplement 2.** Sample size computation based on the effect size of the prestimulus and the evoked activity within the significant time-window for the electrical microstimulation.

**Figure supplement 3.** Sample size computation based on the effect size of the prestimulus and the evoked activity within the significant time-window for the mechanical stimulation.

**Figure supplement 4.** Sample size computation based on the prestimulus effect sizes preceding the electrical microstimulation and the mechanical stimulation.

**Figure supplement 5.** Sample size computation based on the effect size of the stimulus voltages (ERPs within the significant time-window after the electrical microstimulation, and after the mechanical stimulation).

percutaneous microstimulation and justified the transition from the acute to implantable interface to pursue a long-term use.

## Experiments with a transradial amputee

The 3AFC psychophysical protocol used during microstimulation experiments was also evaluated during sessions with subject DAS, a transradial amputee that was implanted with TIMEs (*Raspopovic, 2014*). The sensations elicited by temporal modulation of spiking electrical stimulation of the median nerve through a single implanted electrode were reported by DAS as pertaining to the palmar side of the index finger of the missing hand and as providing a realistic representation of the mechanical alternation between ridges and gaps of the experimented gratings (*Video 1*). DAS showed to be able to comply with the entire 3AFC psychophysical protocol via neural stimulation, without any visual input or guidance by the experimenter, and without interruption in each session, and in 77/80 trials (*Figure 1c*, *Figure 9a*) of trials he was able to correctly identify whether the two sides of the surface had the same SP or which one had a larger SP (*Figure 10*). This performance was significant (p<0.05 confidence, Clopper Pearson exact interval) for each presented single surface (*Figure 9b*). These results indicate that the discrimination ability obtained using TIME stimulation was higher than the one with microstimulation (*Figure 1c*, *Figure 3*). This was attributed to the prolonged use/training of the implanted neural interface before the present study (*Raspopovic, 2014*).

## Analysis of neural coding strategies

The behavioral performance as a function of stimulus in subject DAS can be fit exactly (R = 1) using a logistic function of the difference of SPs between the two halves of each surface (*Figure 9c*). To understand how the very good surface discrimination, which was shown both by intact subjects with microstimulation and by DAS with the TIME implant, was achieved (*Figure 1c*), we tested which neural features correlated with the difference of SPs. During the conditions of constant sliding velocity (10 mm/s) and regulated load force (400 mN), the periodic contact between the sensor and the texture ridges generated regular patterns in the sensor output. The spiking response was given by a sequence of bursts of reliable duration, time-locked to the alternation of ridges and grooves during the sliding motion of gratings (*Figure 10*). This suggested that the Inter Burst Interval (IBI, illustrated in *Figure 10a*), given by the temporal distance between the onset of consecutives bursts, might characterize the response to each texture. Indeed, the MNT process led to linear correlation between the stimulus SP of each texture and a highly specific IBI (ANOVA test, Tukey Kramer correction for multiple comparisons, p<0.00001) reproducing neurophysiological findings in non-human (*Darian-Smith et al., 1980a*; *Phillips and Johnson, 1981*) and human (*Oddo et al., 2011b*) primate studies with similar gratings. Consequently, the difference in IBI between the MNT output that was elicited by the two sides of each stimulus correlated almost perfectly ($R^2 = 0.997$) with the difference

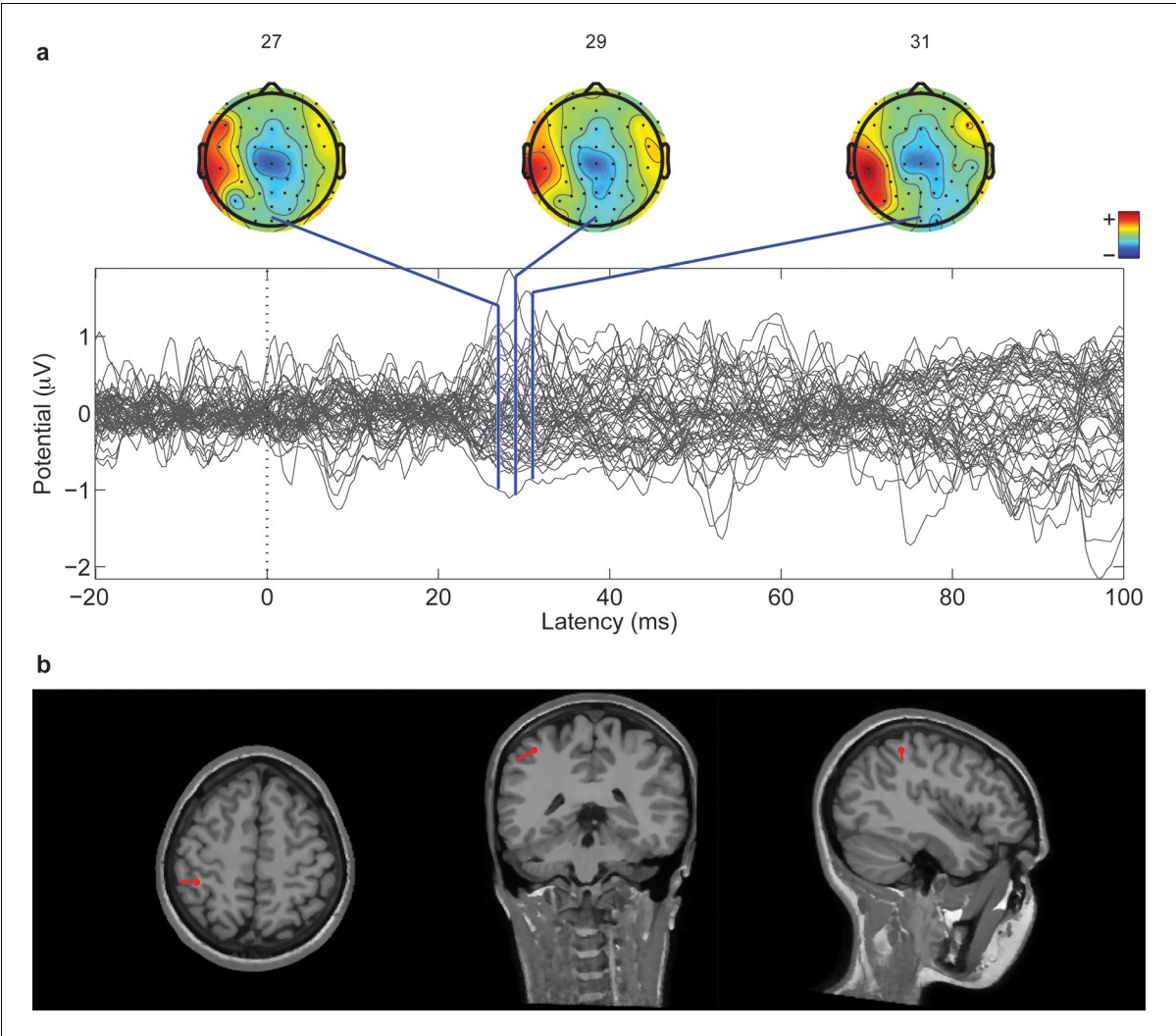

**Figure 5.** Cortical localization of a 1 Hz electrical microstimulation sensory evoked potential. (**a**) Butterfly plot of SEPs for all 64 channels of one subject (M4). All traces are aligned to the electrical stimulus delivery. On top, the topographic representation of amplitude distribution at different time lags (ms). (**b**) Position of the associated equivalent dipole, which corresponds to P27 peak, superimposed on the individual horizontal, coronal and sagittal structural MRI planes of the subject.

in SP (*Figure 9d*). Additionally, the behavioral responses of DAS were perfectly (R = 1) fit by a logistic function of the difference in IBI.

The abovementioned results show that DAS behavioral responses may be based on the temporal structure of stimulation. However, they do not rule out the possibility that the responses were based simply on the count of the total number of spikes injected during each stimulus. Then, we tested whether a rate code not taking into account the temporal structure of the response was able to lead to comparable discrimination of stimuli as the one observed behaviorally. A commonly accepted definition of a rate code is the count of spikes within the time frame of encoding window (*Theunissen and Miller, 1995*; *Panzeri et al., 2010*) which in our case is assumed to be the 2 s presentation of each stimulus. Then, the sensor response to each stimulus, based on which we performed the IBI-based analysis, was also quantified by measuring the overall average firing rate (AFR), i.e., the ratio between the total number of spikes emitted during the sliding and the (fixed) 2 s sliding duration. We found that the AFRs generated in response to different SPs were often similar because a larger IBI was associated with a more intense bursting activity (*Figure 10*). In particular, the AFRs of responses to 0.5, 1.0 and 1.5 mm were not significantly different, and the AFRs of responses to 2.0 mm and 3.0 mm were not significantly different (p<0.05 ANOVA test with multiple

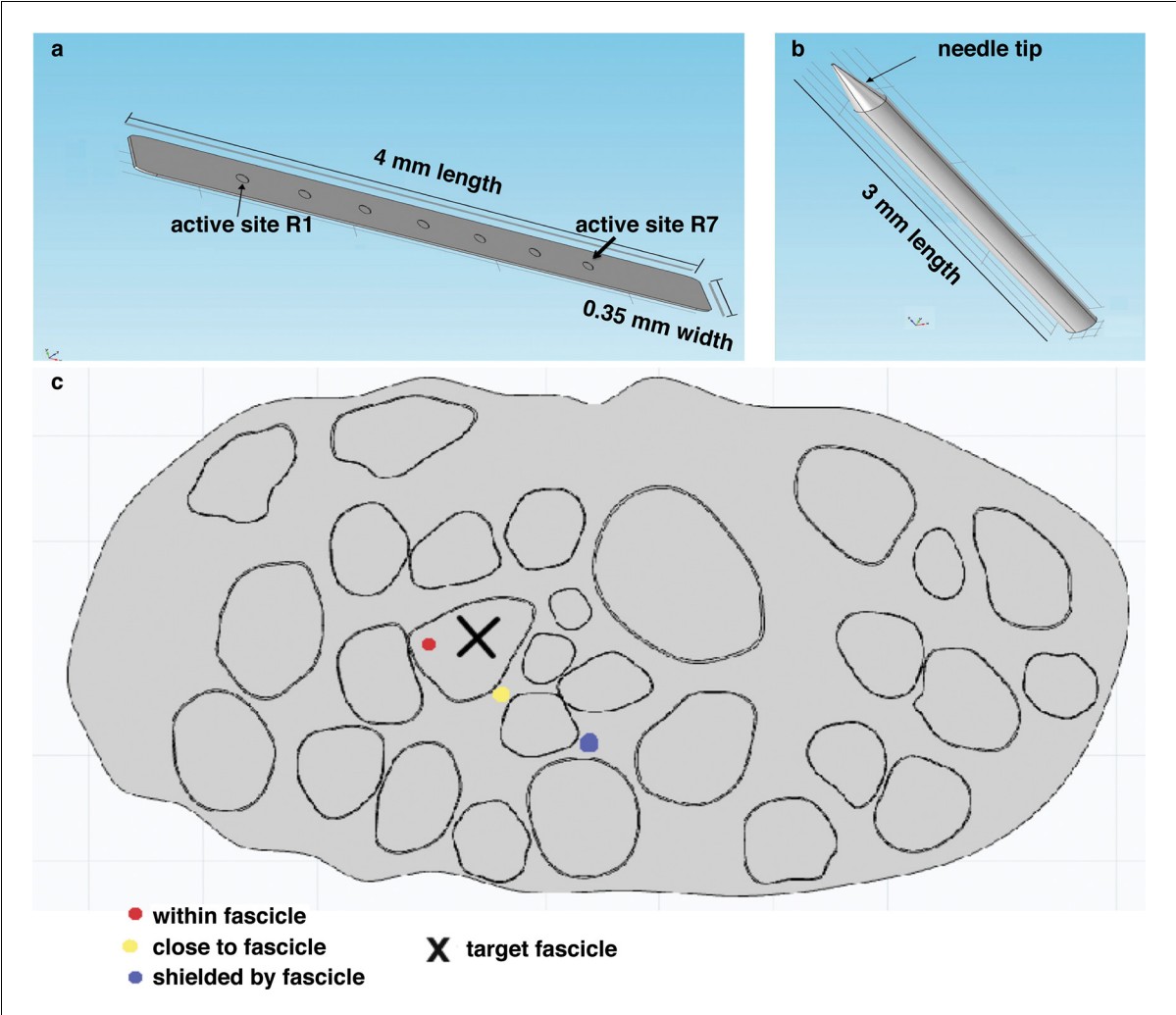

**Figure 6.** Representation of the physical design of electrodes for the hybrid model. (**a**) Implementation for the TIME electrode. (**b**) Implementation for the microneedle. (**c**) Different locations for stimulating active site and tip, that were used in the model to compare the TIME interface versus the microstimulation needle. Red dot represents the evaluated intrafascicular positioning of the TIME active site and the end of microstimulation needle tip. Yellow dot represents close-to-fascicle location, where the end of microstimulation needle tip and the active site of TIME interface were placed. Blue dot represents an example position with a shielding fascicle. The X marker represents the targeted fascicle, where the fiber activation was calculated for different locations of microneedle and TIME for 9 different populations that emulated biologically inherent uncertainty in the placement and extension of fibers innervating a specific hand district. This procedure was performed analogously for medium and small fascicles and confirmed the results.

The following figure supplement is available for figure 6:

**Figure supplement 1.** Finite element model development for the human median nerve, starting from histological pictures and resulting in the solution of voltage distribution within the nerve.

comparisons). Consequently, the difference ΔAFR in AFR between the two sides of a stimulus correlated very weakly with the difference between their SP ($R^2 = 0.066$ for ΔAFR vs $R^2 = 0.997$ for ΔIBI, compare *Figure 9e* and *Figure 9d*). Note that no logistic function of the difference in AFR was able to fit DAS behavioral responses ($R^2 = 0.04$, $p = 0.8$), which was a significant contrast with the perfect fit achieved with ΔIBI. Therefore, because the total spike count was not able to encode differences between stimuli to a degree compatible with behavioral responses, we concluded that the rate code was not responsible for the subject perception of the presented textures. This further strengthens

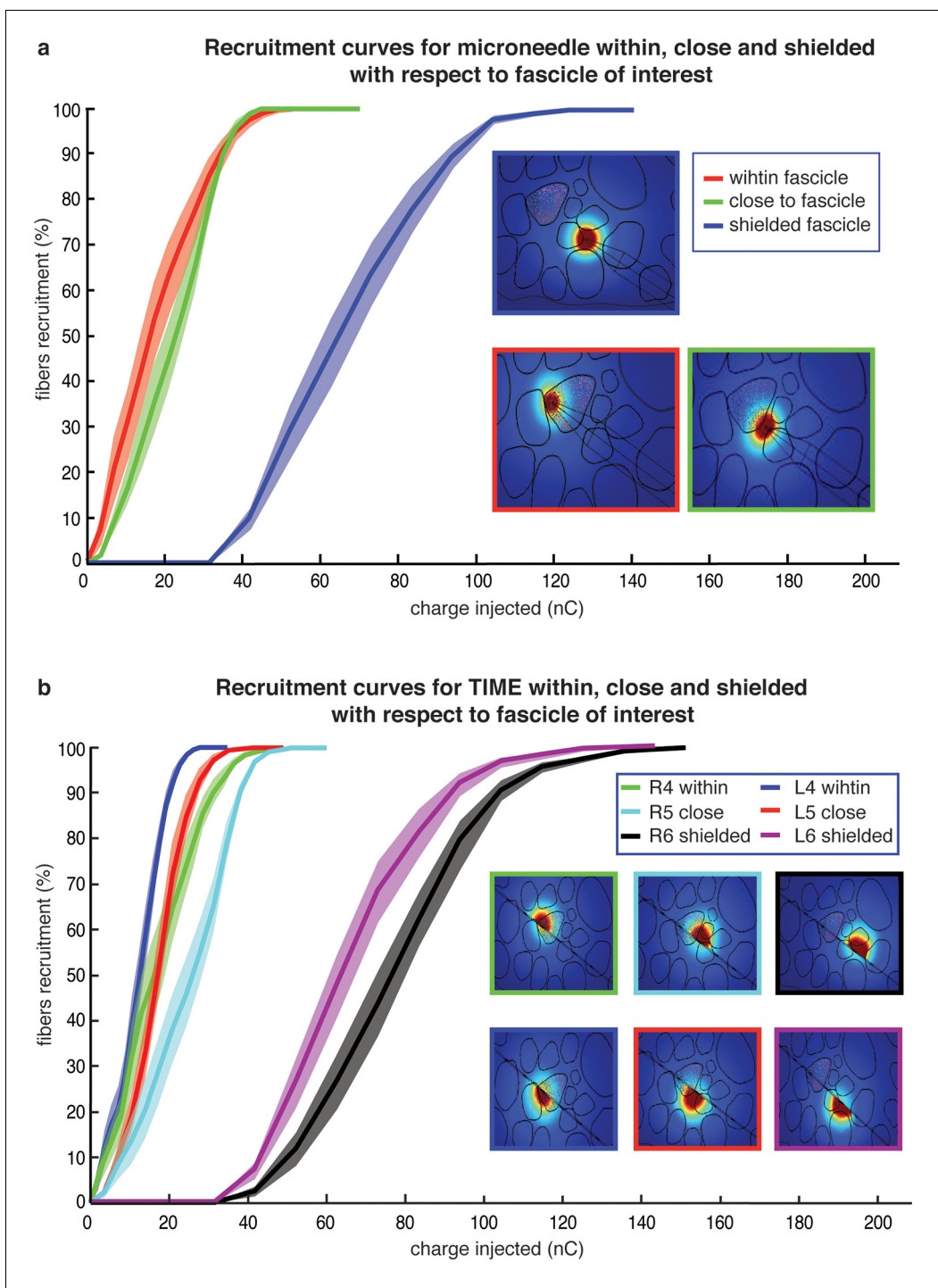

**Figure 7.** Comparison of needle microstimulation and TIME stimulation using hybrid FEM/Neuron models. (**a,b**), Recruitment curves of sensory axon populations resulting from different positions of the active sites for the needle microstimulation and TIME stimulation (mean +/- S.E.M. of percentage of recruited fibers). R4/L4 is the pair of active sites within the target fascicle; R5/L5 is close to it, and R6/L6 is shielded with respect to it. For each position of the stimulating tip/active site, nine different axonal populations were computed. The fiber was considered active when the spike travelled until the last node of Ranvier that was implemented. Figure insets represent voltage distributions for different positions of active sites, as calculated using the FEM solver. These results are representative for several electrode insertion configurations. The same computations were performed for medium and small fascicles and confirmed the results.

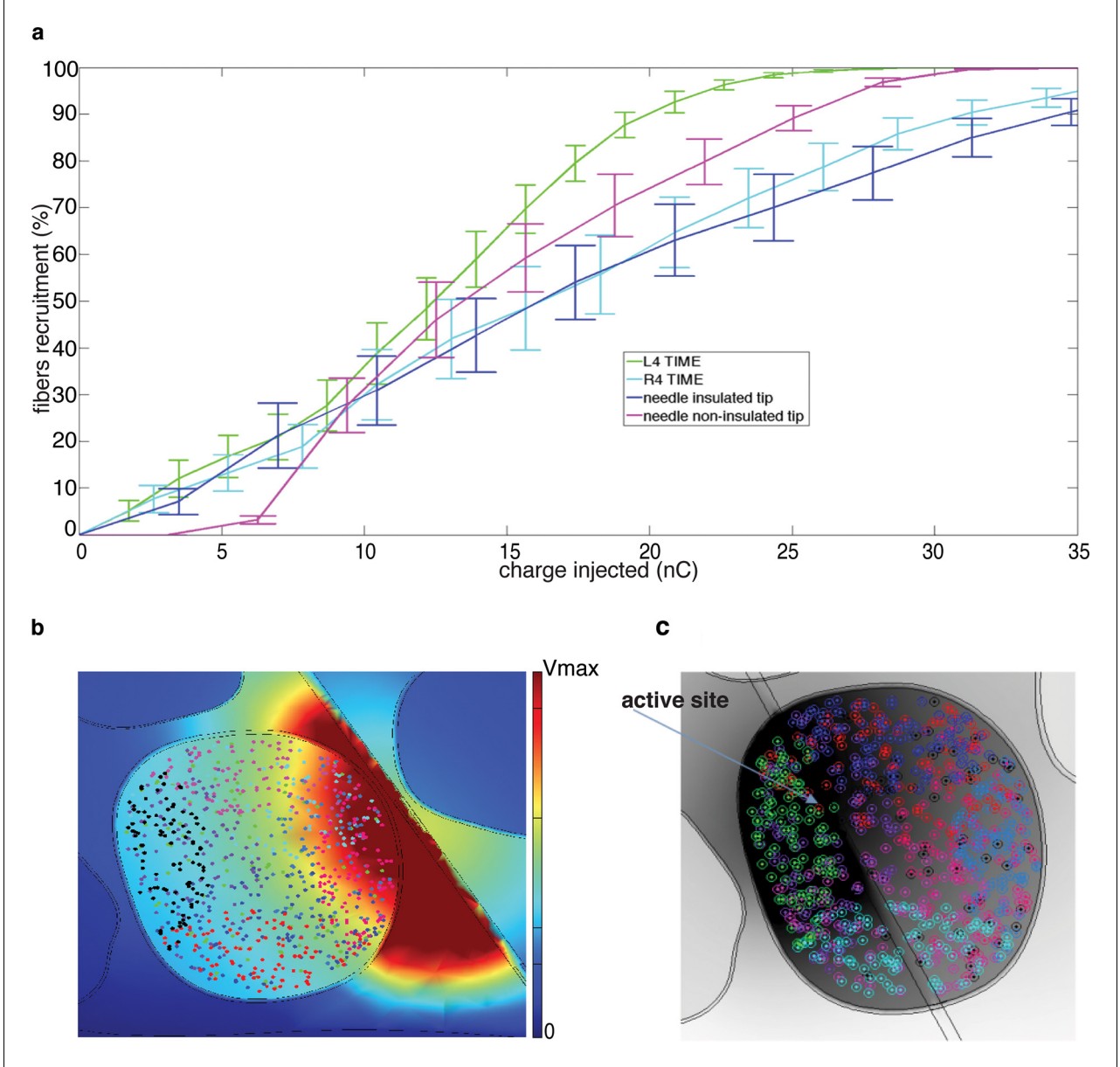

**Figure 8.** Fiber recruitment as a function of injected charge with microstimulation and the TIME implant. (**a**) Recruitment results obtained for placement of the microneedle tip (insulated and non-insulated tip) and the TIME active site (L4 and R4) within the fascicle of interest. The recruitment curves are the mean +/- S.E.M. for the 9 implemented populations in 4 different FEM configurations. The results for the microstimulation needle and TIME interface are similar, which supports the translation from percutaneous towards intraneural stimulation. The 2 cases of microneedle tip exposure yielded comparable outcomes. (**b**) Voltage distribution induced by the close active site and positioning of 9 different fiber populations that were implemented to emulate the inherent anatomical uncertainty of sensory axon locations (see *Figure 8—figure supplement 1* for the correspondence between each color of the fibers illustrated in the panel and the specific implemented population). (**c**) The colored dots indicate another randomized positioning of 9 different fiber populations implemented in the simulations of the hybrid electrical-biophysical model of the median nerve, with active site inside of fascicle.

The following figure supplement is available for figure 8:

**Figure supplement 1.** Representation of the 9 neural populations within large fascicles that were implemented in the hybrid electrical-biophysical model simulations to compare needle microstimulation and stimulation via implanted TIME interface.

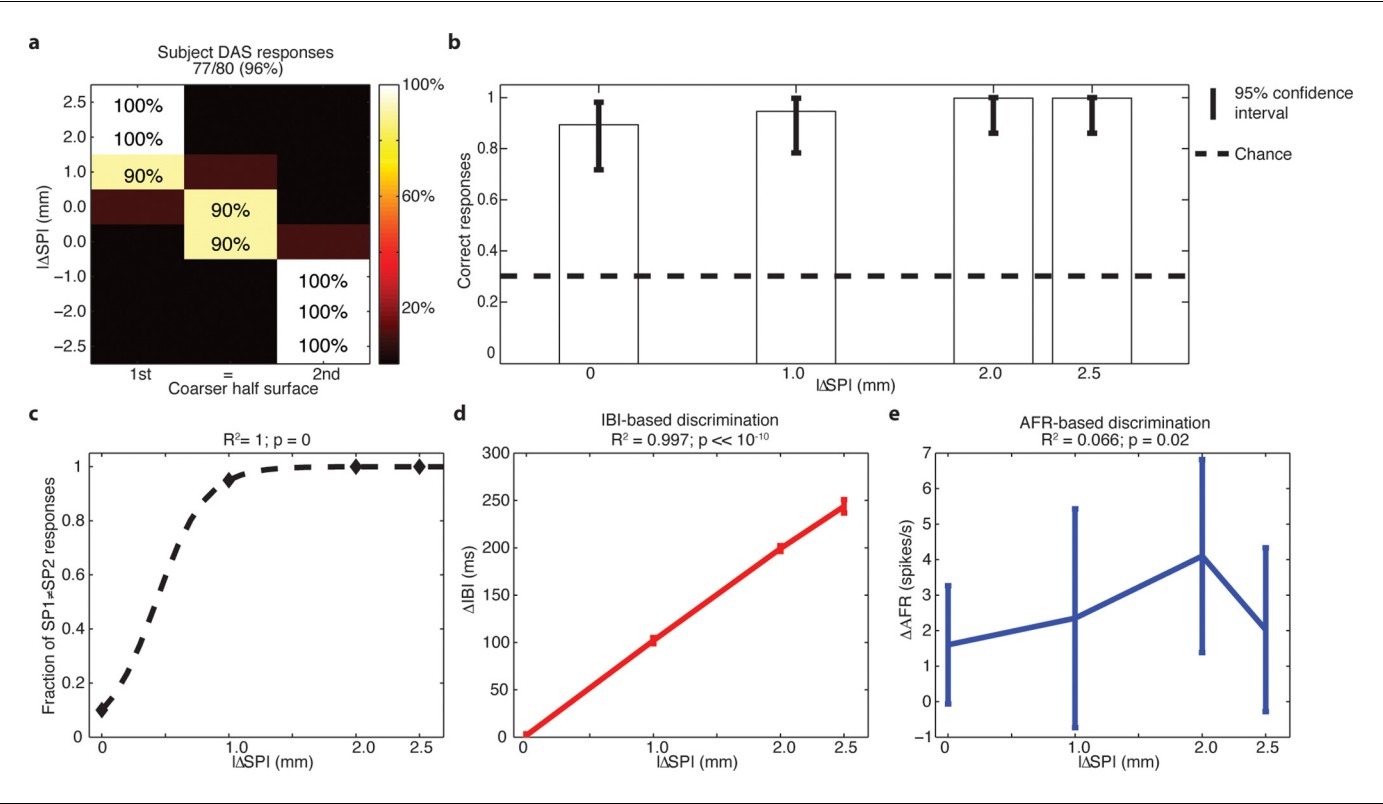

**Figure 9.** Subject behavior and analysis based on the stimulus spatial period (SP), inter-burst interval (IBI) and average firing rate (AFR) in the session with DAS amputee. (**a**) Confusion matrix of the responses given by DAS subject. (**b**) Vertical bars display the correct responses, which are associated with each stimulus. The vertical solid lines over each bar indicate the 95% confidence intervals (Clopper Pearson exact interval) per stimulus. The dashed horizontal line indicates chance level (1/3). (**c**) Fraction of trials for which a pair of stimuli is perceived as different as a function of the difference in spatial period ($\Delta$SP) between the two stimulus halves, and logistic fit (dashed line). The title reports the $R^2$ associated with the fit, i.e., the fraction of data variance explained by the logistic function, and the significance of the Pearson correlation between data points and the fit. (**d,e**) Comparison between IBI-based and AFR-based discrimination. Difference between IBIs ($\Delta$IBI, panel **d**) and difference between AFRs ($\Delta$AFR, panel **e**) measured in the spike patterns elicited by the two sides of each stimulus and plotted as a function of the difference between grating SPs ($\Delta$SP). Error bars indicate the interquartile range. The title reports the fraction of explained variance.

the hypothesis that the subject exploited the temporal structure of the response to discriminate the stimuli.

*Figure 10* shows the instantaneous firing rate (*Lánský et al., 2004*) of different responses computed as the inverse of the inter-spike interval. It is apparent that the temporal evolution of the instantaneous firing rate matches the alternation of ridges and grooves. Indeed, it is already known that, when sliding occurs at fixed velocity, the temporal structure of the sensor output is a linear transformation of the spatial structure of the texture (*Oddo et al., 2011b; 2011c*), at least for regular textures such as those considered here. Therefore, this is a case of temporal coding of the stimulus (*Theunissen and Miller, 1995; Borst and Theunissen, 1999*): the temporal evolution of the spiking response reflects the time evolution of the stimulus rather than internal dynamics of stimulus encoding. Because the spatial structure is highly regular and the textures primarily differ for a single spatial variable (the SP), it is possible to discriminate between them using the corresponding single temporal variable (the IBI). Then, this may be the feature of the injected spike trains that allows subjects to precisely decode the presented stimuli.

## Discussion

Recent neuroprosthetic studies showed that pressure sensation can be elicited by injecting a train of pulses with a fixed shape and different frequencies via intrafascicular (*Dhillon and Horch, 2005*) or

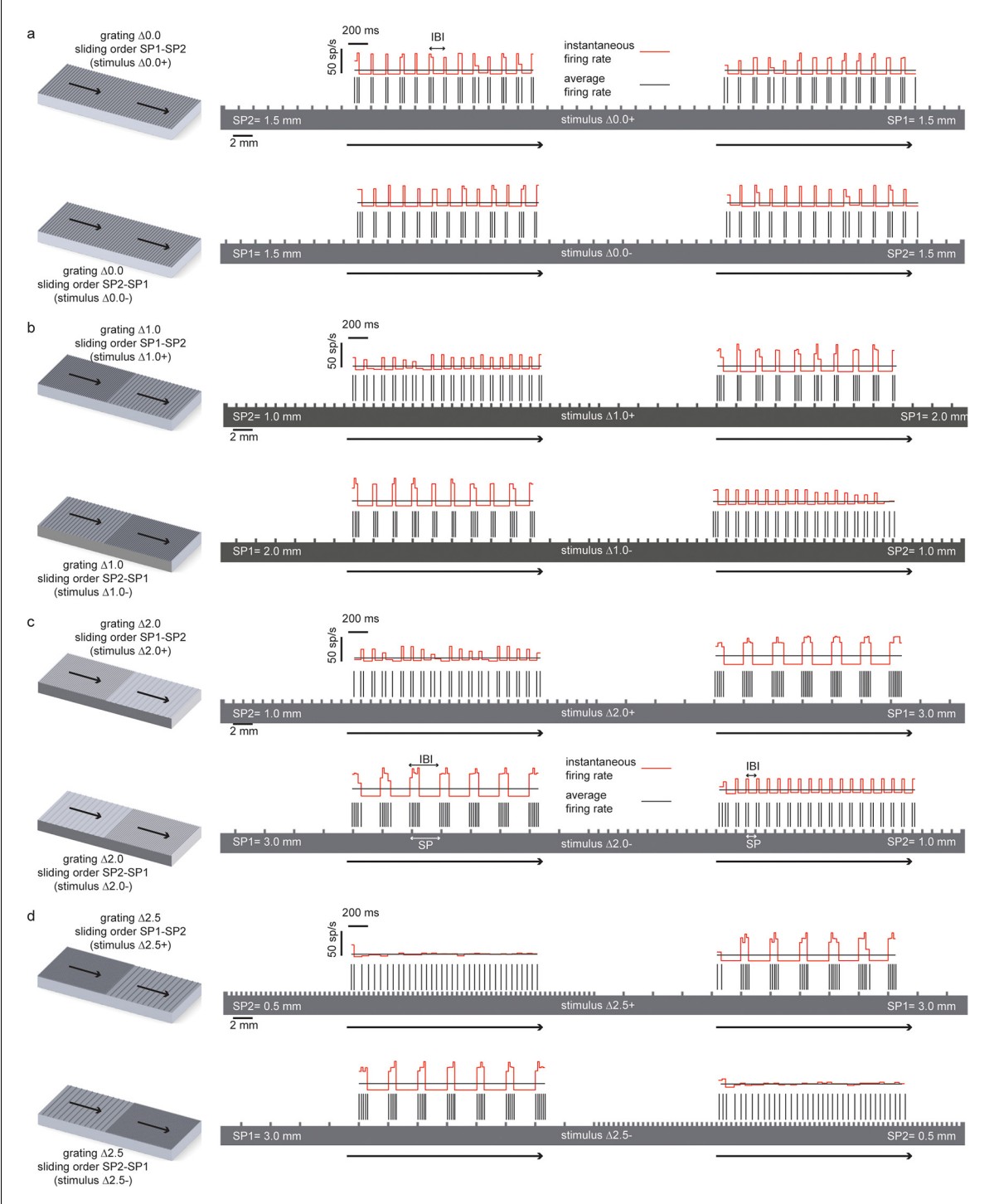

**Figure 10.** Temporal coding of the spatial features of the experimented tactile stimuli. In all panels the spatial structure (scale: 2 mm) of the grating is superimposed over a sample temporal pattern (scale: 200 ms) of the spike train that was obtained as a result of transduction with the artificial finger. The ratio between the spatial and temporal scales turns into the stimulus sliding velocity (10 mm/s). The instantaneous firing rate (scale: 50 spikes/s) is shown together with the average firing rate during the stimulus sliding motion. (a) (above) Stimulus Δ0.0+ is characterized by thw presentation of two half surfaces in the SP1 - SP2 order (first: 1.5 mm, second: 1.5 mm); (below) stimulus Δ0.0-, is characterized by the presentation of two half surfaces in the SP2 – SP1 order (first: 1.5 mm, second: 1.5 mm). For stimulus Δ0.0, the two half surfaces have the same coarseness. Therefore, Δ0.0+ and Δ0.0- result in spiking patterns with a common temporal structure. (b), (above) Stimulus Δ1.0+ is characterized by the presentation of two half surfaces in the SP1 - SP2 order (first: 2.0 mm, second: 1.0 mm); (below) stimulus Δ1.0- is characterized by the presentation of two half surfaces in the SP2 – SP1 order (first: 1.0 mm, second: 2.0 mm). (c) (above) Stimulus Δ2.0+ is characterized by the presentation of two half surfaces in the SP1 - SP2 order (first: 3.0 mm,

*Figure 10 continued on next page*

*Figure 10 continued*

second: 1.0 mm); (below) stimulus Δ2.0- is characterized by the presentation of two half surfaces in the SP2 – SP1 order (first: 1.0 mm, second: 3.0 mm). (**d**) (above) Stimulus Δ2.5+ is characterized by the presentation of two half surfaces in the SP1 - SP2 order (first: 3.0 mm, second: 0.5 mm); (below) stimulus Δ2.5- is characterized by the presentation of two half surfaces in the SP2 – SP1 order (first: 0.5 mm, second: 3.0 mm).

The following figure supplement is available for figure 10:

**Figure supplement 1.** Spatial modulation index that was calculated from the artificial-touch spike patterns from our study (left) compared to neurophysiological data shown by Phillips and Johnson for SA1 units (right, adapted from a previous study (*Phillips and Johnson, 1981*).

---

cuff (*Tan et al., 2014*; *Ortiz-Catalan et al., 2014*) interfaces. Additionally, reconstruction of tactile sensation of force levels and object shape was shown using multi-electrode stimulation via pulses with current amplitude linearly increasing with sensor outputs (*Raspopovic, 2014*). In contrast, this work focused on eliciting textural features by injecting a biomimetic (*Saal and Bensmaia, 2015*) temporal structure of pulse trains with a fixed current amplitude in each experimental session. Discrimination of textural features is a remarkable skill of our somatosensory system, which is used in everyday activities to interact with a peri-personal space. The subjects that use hand prostheses will significantly benefit from the restoration of this sensory function, which represents the next desirable feature after recent achievements (*Tan et al., 2014*; *Raspopovic, 2014*; *Ortiz-Catalan et al., 2014*). The results of this study show that the texture discrimination skills can be artificially provided to users using both needle microstimulation in intact subjects and implantable intraneural interfaces in a transradial amputee. Both in needle and TIME-based experiments, the subjects were able to use our 'artificial' feedback to perform a 3AFC psychophysical test with very good and comparable performance.

The similarity between the needle and TIME-based stimulations was predicted by our hybrid FEM-biophysical model, which supported the transition from acute preparation towards chronically implanted electrodes. In addition, this result opens up interesting scientific and clinical opportunities. In the future, needle microstimulation can be used during translational experiments to test different types of peripheral neural coding strategies, which - if successful in this prototypal situation - would be implemented using implantable neural interfaces in disabled patients. Furthermore, model predictions were confirmed by the quasi-perfect discrimination (96% overall stimuli, see *Figure 1c*, and see *Figure 9a,b* for the analysis on a per-stimulus basis) that was achieved by DAS amputee with TIME stimulation.

The results of the present study showed that a temporal neuronal coding of spatial structure (*Weber et al., 2013*) can successfully elicit tactile cues in case of coarse surfaces that were explored under a passive- dynamic-touch protocol with a constant sliding velocity and a regulated load force, thus, involving controlled motion of the tactile stimulus while the fingertip was not moving (*Yoshioka et al., 2001*). In such passive-touch framework there is a lack of voluntary movement. This allows decoupling of cutaneous information (which was our interest in this study) from the kinaesthetic afferent sensory feedback and efference copy associated with movement dynamics. Thus, it was possible to compare the EEG activities evoked by the substitutive neuromorphic electrical stimulation versus the natural mechanical tactile stimulation. No significant differences in topography or frequency modulation clustering were shown by the EEG signals in the two cases.

The behavioral results showed that injecting the sensor output via a single stimulation channel was sufficient to induce accurate responses (particularly from DAS) in the designed experimental protocol with locked indentation force and tangential velocity of stimulus pairs. Additionally, these experiments allowed us to investigate how the subjects were able to achieve these results in this specific case. In fact, the relationship between IBI and behavioral performance seems to indicate that responses of subject DAS were based on temporal structures of the injected stimuli (which, due to the linearity of our MNT process in mapping the geometry of stimuli, correspond to temporal modulation in the spiking activity captured by the IBI for regular textures), rather than on their average firing rate alone. IBI was previously shown to be given by the ratio between the surface spatial period SP and the sliding velocity V in human (*Oddo et al., 2011b*) and animal (*Weber et al., 2013*; *Darian-Smith et al., 1980a*) neurophysiological investigations and in artificial touch (*Oddo et al., 2011c*) studies. Because in our protocol the sliding velocity V was fixed (10 mm/s), the IBI was

proportional to the SP, and so were their differences (ΔIBI and ΔSP) between the two sides of the stimuli (*Figure 9d*).

Our results indicate that a temporal single-channel MNT-based intraneural stimulation allows gathering of textural features of medium-coarse surfaces with constant and slow sliding velocity. However, for more complex texture discriminating tasks it is plausible that integration of spatiotemporal information from neighboring receptors distributed along the fingertip (*Johansson and Flanagan, 2009*; *Jörntell et al., 2014*) and, thus, multi-channel stimulation, combined with amplitude modulation (*Raspopovic, 2014*), would be needed in less restrictive conditions that involve velocity or force variations and everyday life stimuli (*Weber et al., 2013*; *Dépeault et al., 2008*; *Rongala et al., 2015*).

The promising results obtained with microstimulation in four intact subjects, combined with robust translational indications from the hybrid model and an excellent outcome from one amputee, prompt the idea that neuromorphic stimulation could be a natural and effective tool for eliciting texture discrimination abilities via hand prostheses. Neuroprosthetic research (*Tan et al., 2014*; *Raspopovic, 2014*; *Dhillon and Horch, 2005*; *Rossini et al., 2010*) can in parallel contribute to the evaluation of open neuroscientific hypotheses about tactile perception (*Weber et al., 2013*; *Yoshioka et al., 2001*; *Johansson and Flanagan, 2009*; *Hollins and Risner, 2000*) for the advancement of understanding of human somatosensory physiology.

## Materials and methods

### Sensorized finger

The core element of the artificial fingertip was a Micro Electro Mechanical System (MEMS) sensor with 4 transducing piezoresistors implanted at the base of a cross-shaped structure (*Beccai et al., 2005*) (*Figure 3*). The MEMS was packaged with polymeric compliant material (Dragon Skin, Smooth-On, USA). Sensor data were sampled at 380 Hz per channel by a 24-bit Analog to Digital Converter (ADS1258, Texas Instruments, USA) integrated in the fingertip, and acquired via SPI by a Field Programmable Gate Array (Cyclone II FPGA, Altera, USA).

### Mechano-neuro-transduction (MNT) process

The FPGA streamed the acquired information via Ethernet to a PC for real-time implementation of an artificial mechanoreceptor model emulating the tactile coding recorded during human microneurography sessions carried out with surfaces and experimental apparatus as those of current study (*Oddo et al., 2011b*; *2011d*). To this aim, MEMS sensor signals were converted into sequences of neural spikes via a real-time C++ implementation of the Izhikevich spiking neuron (*Izhikevich, 2003*). Such model was originally conceived to emulate neuron-to-neuron signaling, whereas in this modified implementation the input is haptic rather than synaptic. As detailed in the following sections, these transduced spike trains were injected in the median nerve as train of pulses of fixed width and amplitude through microstimulation electrode in the case of intact subjects and through TIME interface in the case of DAS amputee.

We did not model the fine details of the complex spatio-temporal mechanical interaction between physical stimuli, soft tissues and receptors (*Hayward et al., 2014*), but we injected as input current in the neuron model a signal generated from the outputs of sensors integrated in the artificial fingertip, as follows. Sensor piezoresistive outputs ($S_{x+}$ and $S_{x-}$, represented in blue and in green in *Figure 2c*) belonging to opposite tethers of the cross-shaped structure were subtracted (*Equation 1*) to obtain a component, proximal to distal on the biomimetic fingertip, highly correlated to the frictional shear stress (*Oddo et al., 2007*) arising along the direction of the stimulus sliding motion. This component was half-rectified and amplified (*Equation 2*, as represented in red in *Figure 2c*) in order to inject it as input current in the spiking neuron model ($I_x$ in *Equation 3*). *Equations 3 and 4* describe the subthreshold evolution of the membrane potential $v$ and the recovery variable $u$ in the implemented artificial neuron model (*Spigler et al., 2012*; *Izhikevich, 2003*). Whenever the membrane potential $v$ reached the threshold level, a spike was triggered, $v$ was set to a reset value $c$ and $u$ was increased of a fixed value $d$ (*Equation 5*). The spike was broadcasted by TCP communication to successive services and graphical user interface (Labview, NI, USA).

The computed binary response constituted the output of the neuromorphic artificial touch system, which triggered via USB the current stimulator (STG4008, Multichannel System, Germany) each time that a spike template had to be delivered.

The coefficients of the model (*Spigler et al., 2012*; *Izhikevich, 2003*) were tuned so to achieve a phasic firing with respect to the moving ridges of surfaces (*Gardner and Palmer, 1989*). The MNT process emulated to some extent the neural representation of stimulus spatial patterning observed in previous non-human (*Weber et al., 2013*; *Darian-Smith et al., 1980a*) and human (*Oddo et al., 2011b*) primate recordings of SA1 afferents associated to Merkel mechanoreceptors, a functionally relevant class of tactile units able to encode medium-coarse textures (*Weber et al., 2013*; *Phillips and Johnson, 1981*; *Yoshioka et al., 2001*; *Darian-Smith et al., 1980b*) such as the surfaces evaluated in the present study. The MNT process coded the temporal period of the surface, a feature scaling linearly with spatial period of the grating and inversely with the stimulus tangential velocity. The resulting firing (*Figure 2*) was characterized by biomimetic (*Sathian et al., 1989*) bursts of multiple spikes per ridge with coarser surfaces (see 3.0 mm and 2.0 mm SP in *Figure 10*), by triplets or duplets for surfaces with medium coarseness (see 1.5 mm and 1.0 mm SP in *Figure 10*), down to a single spike per ridge with finer surfaces (see 0.5 mm SP in *Figure 10*), thus implementing a neural code of stimulus geometry based on the modulation of discharge rather than on the mean discharge frequency, resembling SA1 electrophysiological recordings under similar tactile stimulation conditions (*Darian-Smith et al., 1980a*; *Phillips and Johnson, 1981*) (*Figure 10—figure supplement 1*). Furthermore, the instantaneous firing rate of the implemented artificial mechanoreceptor model achieves a modulation up to tens of spikes/s (see red traces in *Figure 10*), coherently with typical values reported in the literature for SA1 units (*Johnson, 2001*; *Phillips and Johnson, 1981*).

$$S_x = S_{x+} - S_{x-} \tag{1}$$

$$I_x = \left\{ \begin{array}{l} KS_x, x \geq 0 \\ 0, x < 0 \end{array} \right\} \tag{2}$$

$$\frac{dv}{dt} = Av^2 + Bv + C - u + \frac{I_x}{RC_m} \tag{3}$$

$$\frac{du}{dt} = a(bv - u) \tag{4}$$

$$if(v \geq v_{th}), then \left\{ \begin{array}{l} v \leftarrow c \\ u \leftarrow u + d \end{array} \right. \tag{5}$$

The following parameters were used: K = 15,000; A = 0.04/sV; B = 5/s; C = 140 V/s; $C_m$ =1F; R = 1C; a = 0.02; b = 0.2/s; c = -65 mV; d = 8 mV; $v_{th}$ = 30 mV.

## Percutaneous electrical microstimulation of the median nerve with intact subjects

Four intact subjects (2 males and 2 females, 25–26 years old) underwent the experiments of texture discrimination with tactile feedback elicited by a stimulation injected through two microneurographic electrodes (FHC UNP40GAS, shaped as needles). The enrolment was subjected to signing a written Informed Consent, approved by the Campus Bio-Medico University Ethics Committee, where this set of experiments took place.

A neurologist trained in microneurography performed the procedure (*Hagbarth and Vallbo, 1967*) , consisting of two phases of electrical stimulation, superficial and percutaneous, necessary to identify the correct site for electrodes insertion. The reference electrode was positioned just through the skin and the active one into the nerve.

Initially, we stimulated the skin of the right arm of the subject in an area centred 2–3 cm proximal and medial to the elbow. A 1 Hz cathodic biphasic balanced square current was delivered to the subject. Pulse duration was 0.2 ms, while amplitude was changed in the range 1–10 mA. The nerve area was identified whenever a hand muscle twitch occurred in correspondence of 1–2 mA amplitude. Reference and active electrodes were then inserted through subject's skin, after cleaning.

A cathodic biphasic balanced square wave was injected through the needles with a frequency of 1 Hz. Pulse duration was 0.2 ms, while the current changed within the 1–100 μA range. The neurologist moved slightly the active electrode towards the inner of the arm seeking for a reported

sensation or a muscular twitch over/of the hand. If one of those conditions occurred with amplitude of the injected current between 1–10 µA, the electrode impaled the nerve fiber. The neurologist then moved the needle in the area of insertion until the subject reported a distinct tactile sensation over one of the first four fingers.

As final evaluation of proper placement of the electrodes, the electroneurogram (ENG) was visualized and acoustically identified: afferent ENG from the median nerve had to be discriminated over the background nerve activity as some mechanical stimuli were exerted over the fingers. Acquisition was performed by means of a system comprising a single channel amplifier (15LT Bipolar, Grass Instrument, USA) and a 16-bit data acquisition board (PCI-6251, NI, USA) installed on a personal computer running a custom interface (Labview, NI, USA).

The subject was instructed to avoid any voluntary movement that could alter the needle placement once the nerve search procedure was completed. An experimenter-guided session of microstimulation was performed in order to identify appropriate parameters to use in the upcoming 3AFC psychophysical protocol: 500 ms trains of cathodic biphasic balanced squares were delivered while changing frequency, pulse duration (Ts) and amplitude (As) starting from 10 Hz, 5 µs and 2 µA, respectively. The tuning of stimulation parameters was operated heuristically among the subjects while performing pilot trials, such as: a) subject connected to the stimulator while the experimented touched manually the artificial fingertip, b) subject exploring the gratings with the non-stimulated hand and then c) perceiving the same surfaces via the artificial fingertip. An additional experimental session of recording somatosensory evoked potentials (SEP) (*Kunesch et al., 1995*) at 1 Hz stimulation rate was performed with subjects M3 and M4 before starting the 3AFC psychophysical experimental protocol: the spike template was injected in the nerve through microstimulation. Stimulation was delivered as square pulses lasting 200 µs at 1 Hz with a number of repetitions (180 for M3 and 700 for M4) allowing a robust Independent Component Analysis (*Artoni et al., 2014*) on EEG data (as detailed in the specific session). In order to allow a better comparison among subjects, stimulation amplitude was individually tailored and chosen as an amplitude that was clearly perceived by the subject, without producing any discomfort (between 5 and 10 µA). Pilot trials of the 3AFC psychophysical test with neuromorphic sensory feedback were then repeated for variable duration among subjects just before starting the actual experimental protocol.

## Intraneural stimulation of the median nerve with implanted interface in transradial amputee

Participant DAS had suffered a traumatic transradial left arm amputation 10 years before the experiments. He was selected from a group of 31 persons with hand amputation because of the stump characteristics and his psychophysical abilities. All procedures were approved by the Institutional Ethics Committees of Policlinic A. Gemelli at Catholic University, where the surgery was performed, IRCCS S. Raffaele Pisana (Rome), where the experiments were performed, the Ethics Committee of Campus Bio-Medico University and the Italian Ministry of Health.

TIME neural interfaces were implanted under general anaesthesia. After superficial disinfection of the medial aspect of the left upper limb, placed extrarotated, the skin was cut along the medial edge of the biceps muscle for about 15 centimetres, from few centimetres below the axilla to about 6 cm above the elbow. The ulnar and the medial nerves were exposed along their course, through careful smooth dissection of the dermal and hypodermic structures, fascial bands, and muscles. Following an epineurial microdissection, performed under a surgical microscope (Zeiss, Pentero) to visualize the fascicles, two TIME electrodes were inserted into the nerve trunk of each nerve (median and ulnar, only the former of which was used for electrical stimulation in the present work). Electrodes were pulled inside the nerve, until the embedded active contacts reached the targeted location, close in contact with the nerve fascicles. Cable segments were placed in subcutaneous pockets, while four holes were made in close proximity of the skin incision, two medially and two laterally for the electrode emission (*Raspopovic, 2014*; *Di Pino et al., 2014*).

The electrodes were removed after 30 days, in accordance with EU guidelines. At the time of removal however, the TIME electrodes were still performing extremely well, and did not cause any discomfort to the subject. The follow-up of the clinical condition of the participant almost 2 years after the end of the protocol did not reveal any subjective or objective side effects.

Upon connection of the neural interface implanted in the median nerve with the current stimulator, an experimenter-guided session of intraneural stimulation was performed in order to identify

appropriate parameters to use in the upcoming 3AFC psychophysical protocol, also capitalizing on the information gathered during the previous days of experimental activities with DAS. Pulse duration (Ts) and amplitude (As) were initially set to 100 µs and 100 µA, respectively. The tuning of stimulation parameters was operated heuristically in 33 min while performing pilot trials, such as: a) DAS connected to the stimulator while he (with the intact hand, to have a sort of self-touch experience) or the experimenter touched manually the artificial fingertip, b) DAS exploring the gratings with the intact hand and then c) perceiving the same surfaces via the artificial fingertip. Pilot trials of the 3AFC psychophysical test with neuromorphic sensory feedback were then repeated for 8 min with the selected parameters (As = 160 µA, Ts = 100 µs) just before starting the experimental protocol.

## Three-alternative forced-choice (3AFC) psychophysical protocol

The protocol structure was essentially the same during microstimulation of intact subjects, and stimulation through implanted TIMEs in the amputee.

The artificial fingertip underwent mechanical stimulation with surface pairs that were presented using a mechatronic platform (*Oddo et al., 2011d*) designed to implement standardized human and artificial passive-touch (*Jones and Smith, 2014*) experimental protocols. The tactile stimuli were gratings, fabricated with 3D printing of plastic material (Project HD 3000, 3D Systems), consisting of a sequence of alternating ridges and grooves with spatial period (SP) from a minimum of 0.5 mm to a maximum of 3.0 mm depending on the stimulus and on the half portion of the surface under test (*Figure 1b*, *Figure 2*, *Figure 10*). The sequence of presented stimuli was randomized within sessions of 16 trials composed of 4 presentations of the 4 surfaces Δ0.0, Δ1.0, Δ2.0 and Δ2.5. In each session, the 4 trials per each stimulus were composed of 2 trials with presentation of the two surface halves in the SP1-SP2 order (labeled +: Δ0.0+, Δ1.0+, Δ2.0+ and Δ2.5+), and 2 trials with reversed SP2-SP1 order (labeled 2: Δ0.0-, Δ1.0-, Δ2.0- and Δ2.5-). The first half-surface was indented at 400 mN on the artificial fingertip and, after 4 s of indentation without tangential movement, it was slid at 10 mm/s for 2 s under regulated load force. The sliding motion was followed by 2 s of indentation at 400 mN without stimulus movement, and then the surface was detached. The same sequence was applied to the second half-surface, starting 3 s after the end of presentation of the first half (see *Figure 2* and *Video 1* for the representation of the whole stimulation sequence). At the end of the presentation of the surface pair, the subject stated whether the first half-surface was perceived as having coarser, finer or same spatial coarseness in comparison to the second half. During the experiment the subject received no feedback about the correctness of the responses.

For the intact subjects who participated in the microstimulation sessions, the gratings were also presented mechanically, via the tactile stimulation platform (*Oddo et al., 2011d*) , directly on the finger (fixed to the platform via a finger holder glued to the nail) that was identified as source of the sensory perception elicited with microstimulation. During the experiment the subjects received no feedback about the correctness of the responses.

The 95% confidence intervals for performance estimate from individual subjects responses were computed with exact Clopper Pearson method (binofit function in Matlab) and compared against chance level (1/3) to assess performance significance. Parameters for logistic fit of performance as a function of stimulus features were estimated with generalized linear regression (glmfit function in Matlab), then the optimal logistic fit function was generated (glmval function in Matlab) and its accuracy evaluated as the squared correlation coefficient between data and fit.

## EEG signals recording

In order to investigate the neural correlates of natural and substitutive texture discrimination, EEG signals were recorded in all microstimulation sessions using a 64 channel EEG device (SD LTM Express, Micromed S.p.A., Italy) with a 2 kHz sampling rate. The montage was in accordance with the 5% 10/20 system (*Oostenveld and Praamstra, 2001*). Careful scalp preparation granted electrodes impedance below 5 kΩ in at least 90% of derivations, as measured at the experiment onset.

## EEG signal processing

Data were analyzed by Matlab scripts based on the EEGLAB toolbox (*Delorme and Makeig, 2004*). EEG signals were processed via independent component analysis (ICA) filtering to remove non-neural sources and artifacts. In order to optimize the dipolarity of the independent components (IC)

extracted (*Artoni et al., 2014*) and to maximize the amount of data fed to the algorithm (while maintaining a sufficient density of information) the EEG continuous data were epoched.

Data were high-pass filtered using a zero-phase 1 Hz, 24[th] order, Chebyshev type II filter and low-pass filtered using a zero-phase, 45 Hz, 71[th] order, Chebyshev type II filter to remove slow drifts and high-frequency noise respectively, then resampled at 256 Hz. Channels with prolonged prominent artifacts (by visual inspection) or with probability more than five times the standard deviation from the mean across all channels were removed (in the end the remaining numbers of channels were 21, 40, 59, 61 respectively for the four subjects), then a common average reference was used for the remaining channels. Epochs containing high-amplitude artefactual potentials, high-frequency muscle noise and other irregular artifacts, as identified by visual inspection, were removed and remaining data were submitted to AMICA (*Palmer et al., 2007*), a generalization of the Infomax algorithm (*Makeig et al., 1996*) to multiple mixture approaches (*Lee et al., 1999*) under the hypothesis that the ICs are spatially static (general stationarity, e.g. recording environment). ICA decompositions were performed separately on each subject over all conditions. Stereotyped artifacts such as eye movements, eye blinks and muscle tension were removed by ICA. The ICA decomposition was then saved and reapplied to data, pre-processed using the same described procedure, but high-passed using a 0.5 Hz, 94[th] order, Chebyshev type II filter but using the pre-computed weights. This procedure allowed to efficiently remove artifacts while retaining the low-frequency EEG information.

Regarding source localization, for subject M4 a realistic head model was obtained by means of the NFT toolbox (*Acar and Makeig, 2010*) using a T1-weighted magnetic resonance (MR) image of the subject and a 4-layer model accounting for scalp ($\sigma$ = 0.33), skull ($\sigma$ = 0.0132), brain tissue ($\sigma$ = 0.33), and cerebrospinal fluid ($\sigma$ = 1.79) respectively. Electrodes positions on the head were co-registered with an optoelectronic neuronavigation system SoftAxic (E.M.S. srl, Italy) and aligned with the Montreal Neurological Institute MNI, Canada. The Finite Element Method (FEM) was used for the numerical solution of the forward problem with Boundary Element Method (BEM) meshes as boundaries. Given the potentials distribution across the scalp, the position of a best-fitting single equivalent current dipole (inverse problem) was determined using the (Dipfit toolbox of EEGLAB) (*Oostenveld and Oostendorp, 2002*; *Delorme et al., 2012*).

EEG recording time-locked with 1 Hz intraneural median nerve microstimulation was exploited to evidence the early somatosensory potentials, the components more strictly due to the stimulation. SEP data epochs were selected from 20 ms before to 100 ms after each stimulation onset. Noisy epochs were rejected by careful visual inspection. Similarly to the continuous data, the criteria for epoch removal were the presence of high amplitude artifacts (e.g., Jaw clenching). Source localization was then performed on the most reliable short-latency potential of cortical origin, namely the P27 peak (*Figure 5*).

Event-related potentials (ERPs) were time-locked to the onset of either the electrical microstimulation or of the sliding phase of the mechanical stimulation (*Figure 4*, *Figure 4—figure supplement 1*). ERPs were normalized for the standard deviation of the prestimulus (1000 ms). ERP's statistical significance between conditions (electrical microstimulation vs. mechanical stimulation) was assessed using a Montecarlo statistics with cluster correction for multiple comparisons (triangulation and maxsum as clustered statistics) (*Maris and Oostenveld, 2007*) , adapted from the FieldTrip toolbox (*Oostenveld et al., 2011*). Statistical power of ERPs comparisons between stimulation conditions was computed for dependent groups (GPower, Duesseldorf, Germany) (*Figure 4—figure supplements 2–5*).

## EEG functional connectivity analysis and EEG graph analysis

EEG functional connectivity analysis was performed using the eLORETA exact low-resolution electromagnetic tomography (*Vecchio et al., 2015*; *Pascual-Marqui, 2011*; *Vecchio et al., 2014a*; *2014b*; *2015b*) software.

To obtain a topographic view of the sensorimotor network, brain connectivity was computed with sLORETA/eLORETA software in 7 regions, positioning the center in Brodmann Areas (BAs: 1–7) separately on the left and right hemispheres.

For each subject and for each hemisphere, among the eLORETA current density time series of the 7 Regions of Interest (ROIs), intracortical Lagged Linear Coherence, extracted using a sphere with 19 mm of radius (*Pascual-Marqui, 2011*; *2007*) , was computed between all possible pairs of

the ROIs for each of the five independent EEG frequency bands: delta (2–4 Hz), theta (4–8 Hz), alpha (8–13 Hz), low beta (13–20 Hz) and high beta (20–30 Hz).

A weighted network was built based on the connectivity between ROIs. The nodes of the network were ROIs, and the edges of the network were weighed by the lagged linear coherence values.

The vertices of the network were the estimated cortical sources in the BAs, and the edges were weighted by the Lagged Linear value within each pair of vertices.

The measure considered here was the clustering (C) that characterizes the tendency of the nearest neighbors of a node to be interconnected. The mean clustering coefficient was computed for all nodes of the graph and was then averaged to estimate the tendency of network elements to form local segregated clusters. Finally, to obtain individual normalized relative measures, the values of each mean clustering coefficient were divided by the mean values obtained by their average in all bands of each subject.

Statistical analysis of percent clustering modulation with respect to baseline period was performed with Statistica v.7 software (StatSoft Inc., USA). Greenhouse and Geisser correction was used for the protection against the violation of the sphericity assumption in the repeated measure ANOVA. Besides, statistical significance was determined by 3-way ANOVA followed by Duncan's multiple range test. ANOVA was performed between three factors: stimulation mode (mechanical and electrical; independent variable), hemisphere (left and right), and band (delta, theta, alpha, low beta, high beta).

## Hybrid electrical-biophysical model of the median nerve for the comparison between microstimulation needle and implanted TIME

Hybrid models (*Raspopovic et al., 2011*; *McIntyre et al., 2002*) account for the anisotropy of extracellular conductivity during the calculation of the electrical field induced by the injection of the electrical current into the tissue, present in real nerves, and for the nonlinear response of cells to the extracellular stimulation. Those two aspects were addressed by means of a finite element method (FEM) to solve the voltage distribution generated by injected currents, and by calculating the neuronal dynamics to estimate the axonal response to the electrical stimulation. The volume conductor model implemented via multiphysics FEM (COMSOL Multiphysics, Sweden) can solve Poisson's equation provided by proper boundary conditions. To do so, an anatomically shaped geometrical model of the median nerve (*Jabaley et al., 1980*) was generated by image segmentation (ImageJ, USA). Coordinates of the image segmentation were exported (livelink COMSOL-Matlab) to edit a 3D nerve model (procedure illustrated in *Figure 6—figure supplement 1*). Perineurium had a thickness of 3% of the diameter of the fascicles (*Grinberg et al., 2008*) , and coordinates were interpolated. The nerve diameter had a maximum of 4.6 mm and minimum of 2.3 mm, as obtained from unpublished histological data provided by Dr. Xavier Navarro and from post-mortem human cadavers dissections at Universitat Autonoma de Barcelona, Spain. A cylinder, representing the nerve's outer space, filled with saline, enclosed the nerve. Boundary optimal dimensions of cylinder were found to be 69 mm for the diameter and 140 mm for the height, using convergence calculations (*Raspopovic et al., 2011*). Electrical ground was therefore fixed in this cylindrical boundary of the structure.

On the biophysical side, MRG model (*McIntyre et al., 2002*) was used to model the nerve tactile sensory fibers. This model represents the nonlinear modified Hodgkin-Huxley Equations for the active compartment of the axons (the nodes of Ranvier) and a detailed realistic representation of the myelinated tracts. The model (available in NEURON model repository) is capable of reproducing several experimental aspects of cells dynamics (*McIntyre et al., 2002*). Then, a fiber with 21 segments (*Raspopovic et al., 2011*; *McIntyre et al., 2002*) of nodes of Ranvier was built and extracellular stimulation procedure used to excite the cell. For a fiber of diameter D, a model had internodal spacing L = 100D. A fiber was considered recruited when a generated action potential travelled along its whole length (i.e., reached the last node of Ranvier). The total recruitment was calculated as the portion of the fibers activated for the specific charge injected with respect to the total number of fibers implemented. Furthermore, nodal length was fixed at 1 μm and nodal diameter scaled from a previous study (*McIntyre et al., 2002*). A list of plausible assumptions had to be taken, during the model construction. The sensory axon populations were constructed (NEURON, USA) by using a probabilistic distribution of fibers diameters (*Vallbo and Johansson, 1984*) for different tactile units of the human hand, resulting in two Gaussian distributions which differentiate nociceptive fibers

from fibers responsible for pressure/touch sensation, and the latter was used. One of the assumptions constraining the model was that the stimulation of different fiber types, such as nociceptive fibers, was not induced, at used current range. A total amount of 100 modeled fibers for each fascicle were placed randomly in the specific target fascicle, for several placements (*Figure 8*), as explained in continuation. As the fiber organization within different fascicles in the nerve is unknown, we assumed that fibers within one fascicle innervate the same hand area (*Jabaley et al., 1980*) (i.e., the fibers in one fascicle are innervating a single finger tip and not many of them). Moreover, since there is an inherent anatomical uncertainty in placement and extension of fibers innervating a specific hand district, for every position of stimulating electrode/needle we implemented 9 populations having different extensions and centroids for the same large fascicle (i.e., spanning from having the whole fascicle uniformly populated, to the case of very concentrated population where the fibers are almost touching each-other, *Figure 8* and *Figure 8—figure supplement 1*). Then the analysis has been performed for the range of significant fascicle sizes, defined as median size representatives of three groups: small (1 population implemented), medium (5 populations implemented) and large (9 populations). Different device placements were also studied: within, close, or far from fascicle, finally resulting in n = 90 simulations for TIME and n = 45 simulations cases for microneedle (this is because for every position of needle tip there are 2 corresponding positions of TIME: left and right).

The microstimulation needle (FHC UNP40GAS) was replicated as a cylinder with a cone-like ending (*Figure 6*): the cylinder had 3 mm length, and cone (tip) with semi-angle 12.78° and shank diameter 250 µm. The whole structure was insulated and had electrical conductivity of 6.67 10–14 S/m and the tip was non-insulated with a conductivity of 1.89 $10^7$ S/m. Since in the case of microstimulation needle the tip was un-insulated by the neurologist, imminently before the insertion, the precise un-insulated cone dimension is not known. In order to account for this uncertainty, two models were implemented and the results analyzed for both (*Figure 8*). The TIME electrode was built as a rectangular structure where seven circled active sites were placed in each side of the structure (*Figure 6*). Conductivity of the polyimide substrate was set to 6.67 10–14 S/m. The radius for the active sites was 40 µm and the thickness 300 nm. The overall structure had 4 mm length, 0.35 mm width and 20 µm thickness. The electrical conductivity values used for the FEM were [0.0826i 0.0826j 0.571k] S/m for the endoneurium (*Raspopovic et al., 2011*; *McIntyre et al., 2002*) , 880 µS/m for perineurium, 0.0826 S/m for the epineurium (*Raspopovic et al., 2011*; *McIntyre et al., 2002*) , 2 S/m for saline.

Conductivity of perineurium was recalculated from a previous study (*Weerasuriya et al., 1984*) taking into account the thickness of the perineurium as 3% of diameter of the fascicle and the difference of temperature between frogs and humans. Although TIME electrode was implanted in vivo and no saline placed outside the nerve, this value was used as in previous studies (*Raspopovic et al., 2011*; *McIntyre et al., 2002*).

The similarity between needle and TIME is estimated by comparison of charges necessary for 10% of recruitment of fibers, for respectively 45 and 90 populations computed, by means of Kruskal-Wallis test, with significance level fixed at 0.05.

## Acknowledgements

The Authors are deeply grateful to all the Participants, and in particular to DAS who freely donated six weeks of his life for the advancement of knowledge and for a better future of People with hand amputation. The Authors are also grateful to Prof. Thomas Stieglitz and his team for the development of the TIMEs, to Prof. Eduardo Fernandez for their surgical implantation, and to Prof. Xavier Navarro for providing histological data of the median nerve. The authors also thank Mr. Matteo Moisé for the contributions in the preparation of the artwork.

The research leading to these results has received funding from the European Community's Seventh Framework Programme (FP7/2007-2013) under grant agreements n° 224012 (TIME, Transverse Intrafascicular Multichannel Electrode system), 228844 (NANOBIOTOUCH, Nanoresolved multi-scan investigations of human tactile sensations and tissue engineered nanobiosensor) and 611687 (NEBIAS, NEuro-controlled BIdirectional Artificial upper limb and hand prosthesiS), from the Italian Ministry of Health (NEMESIS project), from the Italian Ministry of Research (PRIN/HandBot, Biomechatronic hand prostheses endowed with bio-inspired tactile perception, bi-directional neural interfaces and distributed sensori-motor control, CUP: B81J12002680008), from the National Institute for Insurance against Industrial Injuries (PPR2, Control of hand prosthesis by invasive neural interfaces),

from the Wyss Center for Bio and Neuroengineering, and from the Swiss National Competence Center in Research in Robotics.

## Additional information

### Competing interests

CMO: Has a patent entitled "Method and apparatus for transmitting tactile sensations to an user" (Italian patent number 0001417070). SR: Has a patent entitled "Bidirectional limb neuroprosthesis" pending (PCT/IB2014/067143). GS: Has a patent entitled "Method and apparatus for transmitting tactile sensations to an user" (Italian patent number 0001417070). FP: Has a patent entitled "Bidirectional limb neuroprosthesis" pending (PCT/IB2014/067143). DC: Has a patent entitled "Method and apparatus for transmitting tactile sensations to an user" (Italian patent number 0001417070). MCC: Has a patent entitled "Method and apparatus for transmitting tactile sensations to an user" (Italian patent number 0001417070). SM: Has a patent entitled "Bidirectional limb neuroprosthesis" pending (PCT/IB2014/067143). The other authors declare that no competing interests exist.

### Funding

| Funder | Grant reference number | Author |
|---|---|---|
| Directorate-General for Communications Networks, Content and Technology | EU Grant CP-FP-INFSO 224012 TIME project | Stanisa Raspopovic<br>Francesco Petrini<br>Federica Giambattistelli<br>Fabrizio Vecchio<br>Francesca Miraglia<br>Giovanni Di Pino<br>Paolo Maria Rossini<br>Silvestro Micera |
| Directorate-General for Communications Networks, Content and Technology | EU Grant FET 611687 NEBIAS Project | Calogero Maria Oddo<br>Stanisa Raspopovic<br>Alberto Mazzoni<br>Domenico Camboni<br>Paolo Maria Rossini<br>Silvestro Micera |
| Directorate-General for Communications Networks, Content and Technology | EU Grant FP7-NMP 228844 NANOBIOTOUCH project | Calogero Maria Oddo<br>Giacomo Spigler<br>Domenico Camboni<br>Maria Chiara Carrozza |
| Ministero della Salute | Italian NEMESIS (Neurocontrolled mechatronic hand prosthesis) | Calogero M Oddo<br>Stanisa Raspopovic<br>Francesco Petrini<br>Paolo M Rossini<br>Silvestro Micera |
| Ministero dell'Istruzione, dell'Università e della Ricerca | Italian project PRIN/ HandBot | Calogero M Oddo<br>Fiorenzo Artoni<br>Alberto Mazzoni<br>Giacomo Spigler<br>Francesco Petrini<br>Federica Giambattistelli<br>Loredana Zollo<br>Giovanni Di Pino<br>Domenico Camboni<br>Eugenio Guglielmelli<br>Silvestro Micera |
| Italian National Institute for Insturance against Industrial Injuries | National project PPR2 (Control of hand prosthesis by invasive neural interfaces) | Loredana Zollo<br>Eugenio Guglielmelli |
| Swiss National Competence Center in Research in Robotics | NCCR Robotics | Stanisa Raspopovic<br>Francesco Petrini<br>Silvestro Micera |
| Wyss Center for Bio and Neuroengineering | ENABLE Project | Silvestro Micera |

The funders had no role in study design, data collection and interpretation, or the decision to submit the work for publication.

## Author contributions

CMO, Designed the study, supervised and conducted the experiments, developed the sensorized finger and the MNT process, supervised system integration, analyzed data, discussed the results and wrote the paper; SR, Co-designed the study, Supervised and conducted experiments, Developed the hybrid electrical-biophysical model, Analyzed data, and wrote the paper; FA, Performed EEG signal processing and wrote the related parts of the paper; AM, Analyzed sensor output data and behavioral data and wrote the paper; GS, Developed the MNT process and collaborated during system integration and during the experiments; FP, Collaborated during system integration and during all the experiments, Conducted intact human experiments and wrote the related parts of the paper, Contributed to the design of the hybrid electrical-biophysical model; FG, Conducted percutaneous microstimulation sessions with intact subjects; FV, FM, Performed EEG functional connectivity analysis and EEG graph analysis and wrote the related parts of the paper; LZ, Collaborated during the preparation and execution of the experiments; GDP, Selected the amputee subject and collaborated during the experiments and to the discussion of EEG data; DC, Collaborated during system integration, Collaborated during the experiments and contributed to the preparation of artwork of the paper; MCC, Collaborated during the preparation of the experiments and developed the sensorized finger; EG, Collaborated during the preparation of the experiments; PMR, Selected the amputee subject and supervised EEG functional connectivity analysis and EEG graph analysis and wrote the related parts of the paper; UF, Was responsible of EEG signals recording, supervised EEG signal processing and wrote the related parts of the paper; SM, Designed the study, Supervised the experiments, Discussed the results, and wrote the paper

## Author ORCIDs

Calogero Maria Oddo, http://orcid.org/0000-0002-1489-5701
Silvestro Micera, http://orcid.org/0000-0003-4396-8217

## Ethics

Human subjects: Participant DAS had suffered a traumatic transradial left arm amputation 10 years before the experiments. He was selected from a group of 31 persons with hand amputation because of the stump characteristics and his psychophysical abilities. All procedures were approved by the Institutional Ethics Committees of Policlinic A. Gemelli at Catholic University, where the surgery was performed, IRCCS S. Raffaele Pisana (Rome), where the experiments were performed, the Ethics Committee of Campus Bio-Medico University and the Italian Ministry of Health. Four intact subjects (2 males and 2 females, 25-26 years old) underwent the experiments of texture discrimination with tactile feedback elicited by a stimulation injected through two microneurographic electrodes (FHC UNP40GAS, shaped as needles). The enrolment was subjected to signing a written Informed Consent, approved by the Campus Bio-Medico University Ethics Committee, where this set of experiments took place.

# Additional files

## Major datasets

The following dataset was generated:

| Author(s) | Year | Dataset title | Dataset URL | Database, license, and accessibility information |
|---|---|---|---|---|
| Oddo CM, Raspopovic S, Artoni F, Mazzoni A, Spigler G, Petrini F, Federica G, Fabrizio V, Francesca M, Loredana Z, Di Pino G, Camboni D, Carrozza MC, Guglielmelli E, Rossini PM, Faraguna U, Micera S | 2015 | Data from Intraneural stimulation elicits discrimination of textural features by artificial fingertip in intact and amputee humans | http://dx.doi.org/10.5061/dryad.630hf | Available at Dryad Digital Repository under a CC0 Public Domain Dedication |

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
