## [Decision Letter]

Thank you for submitting your work entitled "Intraneural stimulation can restore discrimination of textural features by artificial fingertips in humans" for peer review at *eLife*. Your submission has been favorably evaluated by Timothy Behrens (Senior editor), a Reviewing editor, and two reviewers, one of whom is a member of our Board of Reviewing Editors.

The reviewers have discussed the reviews with one another and the Reviewing editor has drafted this decision to help you prepare a revised submission.

Summary:

This paper reports further on the very good pioneering work by these authors (previously reported) regarding the sensory/tactile input into the peripheral nerve. This paper extends the previous work, with some different aspects, although there is also a considerable overlap (which must be explained). Specifically, the authors study the use of NMT and TIME, in volunteer studies, in the study of a transradial amputee and in a simulation investigation as a way to restore tactile sensation.

The main result of neuromorphic stimulation, and neural response are interesting first presentations.

The strength of the paper is the very detailed experimentation, involving microneurography and the sensory input provided in an amputee. These experiments are aided by giving a biomimetic or neuromorphic input. Further, modeling shows how microneurography and TIM electrodes work comparably. This type of modeling coupled to experiments shows that this paper represents thorough research. The cortical response further validates that sensory perception is observed – this is quite interesting and novel as well. Overall, the work has enormous promise for establishing a paradigm for restoring tactile sensation to patients with limb injuries.

Essential revisions:

First, and most importantly, previous work by Raspopovic, more or less lays out the tactile stimulation. Also, the authors have published their sensors work. The new aspect seems to be the biologically plausible model. But the model selected – Izh ikevich is not a biological model of receptors – does not produce receptor-spiking activity and no transduction is built in.

In the subheading “Mechano-neuro-transduction (MNT) process”, sensor to spike output from Izhikevich is not an ideal receptor to neural activity transformation. The authors should point out this simplification – actually a limitation. The periodic rate output is also a significant simplification. The model produces bursts vs single spikes, but it's not clear it is biomimetic. This interpretation also needs some caution.

Second, the paper does not clearly and cleanly present what is novel here – and differentiate from many prior papers by the authors. What is novel – the stimulus pattern? The computational model? Comparison with 4 normal, amputee? Distinguish from the prior work by the same group. Please emphasize this point in your response letter and revised manuscript. The question of novelty will be critical in making a final decision on the publication of this work.

Third, the work presents tactile encoding – in particular coarse + geometric -> shape sensing. This is not clear. Is it only the artifact of their stimulus (grating) or is this the result of perception by the subjects – having tested many stimuli. i.e. is the perceptual output of the given microneurographic access to nerves and penetrating electrodes' access to the fascicles? If so, how specific and representative is it?

Fourth, the statistical analysis of the data should be improved. The text and Figure 1 discuss the results of the study in terms of the overall success rate. Figure 3 shows the breakdown of the responses by subject. There is a conspicuous lack of analysis of the data. The data should be analyzed not by lumping together all of the subjects and reporting an aggregate successful discrimination rate, but rather by individual subject as a function of stimulus type. The coarser the stimulus, the greater the discriminatory power and the weaker the stimulus the less the discriminatory power. There should be an analysis showing that the performance by individual is greater than what would be expected by chance, where chance here is 0.33. If there is an approximate monotone relation among the stimuli then a logistic regression analysis could be performed. The analysis should be conducted using confidence intervals or Bayesian methods and not just simply by reporting p-values. An advantage of a Bayesian hierarchical analysis or a random effects model across subjects is that they provide a formal way of pooling information across subjects. The latter is crucial for this problem because the authors which to establish that they have been able to provide neuromorphically tactile sensation. This statement should be made with an assessment of the accuracy of the discriminatory power of the MNT technique for each subject as a function of stimulus type. It is likely that the results will reveal that more trials will be required to show that there is truly discriminatory capability.

The authors state that they observed no difference in the natural EEG activity and evoked stimulation in source topography, response timing and clustering. How were these assessments made? They involve negative results and require a power analysis to establish their validity.

Although the data are more compelling for the TIME technique applied to the transradial amputee patient, a similar analysis should be performed for these data as well.

The authors wish to establish from their simulation analyses that the response patterns under MNT and TIME are similar if not effectively the same. It is unclear where the uncertainty comes from in these analyses since they are simulation models. Therefore how can statistical assessments of uncertainty and formal statistical inferences be made? Moreover, the authors wish to infer that the stimulation responses of the two modalities do not differ. They report a p-value without a specific statement about the statistical test being used in the analysis. Because the authors wish to establish a negative result of no difference between the two stimulations modalities, they should report a power analysis stating what types and magnitudes of differences their investigations were calibrated to detect.

The authors do not have to develop an ancillary classifier model to understand the performance characteristics of the MNT and the TIME paradigms. If they build statistical models to analyze the data from the subjects that include covariates that has IBI they will get their answer with the same model being used to analyze the data. This approach would obviate the current analyses which lead to multiple comparisons corrections.

[Editors' note: further revisions were requested prior to acceptance, as described below.]

Thank you for resubmitting your work entitled "Intraneural stimulation can restore discrimination of textural features by artificial fingertips in humans" for further consideration at *eLife*. Your revised article has been favorably evaluated by Timothy Behrens (Senior editor), a Reviewing editor, and one reviewer. The manuscript has been improved but there are a few remaining issues that need to be addressed before acceptance, as described below.

General Comments:

This paper has been revised, and more importantly, careful and thoughtful rebuttal to all the questions has been provided. Supplementary figures and additional statistical analysis are also appreciated. Overall, the paper is acceptable. The authors should note a few salient points and revise their paper accordingly (no further review is needed).

This is an outstanding piece of work. The paper has many novel aspects, now better clarified as requested, to point out the texture discrimination capability in humans, different stimuli and temporal coding, comparison with needle vs intrafascicular, etc. The cortical, neural topographic map, and source localization strengthen the paper. The claim about neuromorphic/Izhikevich is now moderated, although the authors still argue that this is representation of tactile receptors, bur arguably this is still a significant approximation based on the Izhikevich model.

The paper's distinction from the previous, Raspopovic paper is now made and is acceptable. The comments made concerning this, from texture to encoding model, are acceptable.

The novelty aspect has been responded to: a) Texture discrimination (but not "restoration" – see below);

b) Hybrid FEM model is very useful and demonstrates TIME's utility and various fiber and recruiting; c) Perceptual studies, both psychophysical and cortical, strengthen the paper.

Suggested Revisions:

1) The claim in the Abstract is still too broad: "Intraneural MNT-based stimulation restored discrimination of textural features, thus enhancing the user's tactile capabilities." The word "restore" is too strong. It implies permanent restoration/recovery. Here a texture perception has been mimicked, and performance has not been "enhanced."

2) Paragraph two, subheading “Analysis of neural coding strategies”: This information pertains to Figure 10). The message that the rate (in e, worse) vs temporal code (in d, good) is getting lost in this paragraph. This is an important observation and clarification would help. The figure caption could be more explanatory too.

3) Paragraph two, subheading “Hybrid electrical-biophysical model of the median nerve for the comparison between microstimulation needle and implanted TIME”: As the fiber organization within different fascicles in the nerve is unknown, we assumed that fibers within the fascicles belong to same functionality.

Please clarify how nociceptive fibers were separated from sensory (e.g. see the comment pertaining to the model).Indeed, this issue or not being able to separate different fiber types is quite critical and must be marked as a limitation.

4) Figure 2: Traces and colors are not well explained.

5) Figure 6 caption: The X marker represents the targeted fascicle where the fiber activation per 9 different populations was calculated. What does this mean (what populations?) for different locations of microneedle and TIME? This procedure is carried out analogously for medium and small fascicles, confirming results. Please could you also clarify the procedure.

6) Figure 8: outcomes. Color-coding is not explained

7) Figure 10: This figure, slightly modified to clarify rate vs temporal, would be much clearer (and certainly this result is important).

---

## [Author Response]

Essential revisions:

First, and most importantly, previous work by Raspopovich, more or less lays out the tactile stimulation. Also, the authors have published their sensors work. The new aspect seems to be the biologically plausible model. But the model selected – Izhi kevich is not a biological model of receptors – does not produce receptor-spiking activity and no transduction is built in.

In the subheading “Mechano-neuro-transduction (MNT) process”, sensor to spike output from Izhikevich is not an ideal receptor to neural activity transformation. The authors should point out this simplification – actually a limitation. The periodic rate output is also a significant simplification. The model produces bursts vs single spikes, but it's not clear it is biomimetic. This interpretation also needs some caution.

We thank the reviewers for this comment, which allows us to clarify how the mechano-neuro-transduction actually works in our experimental design:

1) We modified the Methods section in order to better point out the limitations of our model. We highlighted that our model:

– Is a modified version of the original Izhikevich model, which was conceived to emulate neuron-to-neuron signaling, whereas in our modified implementation its input is haptic rather than synaptic.

– Does not account for the fine details of the complex spatio-temporal mechanical interaction between physical stimuli, soft tissues and receptors (Hayward et al., 2014).

2) Notwithstanding these limitations, as discussed in the revised paper, our implementation captures the fundamental properties of the firing behavior of SA1 afferents associated to Merkel receptors, a functionally relevant class of tactile units able to encode medium-coarse textures (Darian-Smith and Oke, 1980, Phillips and Johnson, 1981, Yoshioka et al., 2001, Weber et al., 2013), such as the surfaces evaluated in the present study. Specifically:

– In the study by Darian-Smith and Oke (e.g., in Figure 6 of such prior work) the spatial structure of moving gratings characterized by a temporal frequency in the order of 20 Hz is encoded by SA1 afferents into a coherent temporal spiking activity with about one spike per ridge of the stimulus (Darian-Smith and Oke, 1980). Coherently with such previous neurophysiological data, in our study stimuli with 0.5 mm spatial period (resulting in a 20 Hz temporal frequency since the sliding velocity is 10 mm/s) induce about one spike per ridge (Figure 9 of our manuscript).

– As shown in our manuscript in the raster plot in Figure 1 and in Figure 9, the encoding of our coarser surfaces (with 1.0 mm, 1.5 mm, 2.0 mm and 3.0 mm spatial period) reflects the physiological tendency of SA1 units to fire more than one spike per ridge for coarser spatial periods, with bursts properly modulated by the surface structure (towards longer IBI), as depicted in Figure 1 and Figure 2 of the previous study previously by Phillips and Johnson (1981).

– To evaluate such evidences in a more quantitative way, we added Figure 9—figure supplement 1, that shows the spatial modulation index calculated on the spike patterns of our study, to be compared with Figure 9 of the previous study by Phillips and Johnson (1981). The modulation index shown in the spike patterns produced by our artificial touch system reflects the trend (monotonically growing with the spatial period) that was previously reported with physiological data.

3) In the revised manuscript we also clarify that the periodicity of the bursts of neural stimulation delivered to the subjects is not inherent to the model, which is able to fire any kind of spike train patterns including the highly irregular ones induced by the contact with complex surfaces. The periodicity is instead a direct consequence of the experimented stimuli, which are periodic gratings and explored with constant velocity. Hence, the spatial structure of gratings is transduced in a temporal sequence that maps the textural/geometrical features of the stimuli. The capability of the implemented model to generalize with daily life naturalistic stimuli (either macroscopically aperiodic, such as latex and velvet, or with a fine periodic structure such as textiles) was recently demonstrated in a fully-robotic artificial touch study (Rongala et al., 2015), and such set of stimuli will be evaluated in the future protocols with amputees and chronically-implanted neural interfaces.

Second, the paper does not clearly and cleanly present what is novel here – and differentiate from many prior papers by the authors. What is novel – the stimulus pattern? The computational model? Comparison with 4 normal, amputee? Distinguish from the prior work by the same group. Please emphasize this point in your response letter and revised manuscript. The question of novelty will be critical in making a final decision on the publication of this work.

The main novelties of this work are hereafter summarized (and better explained in the manuscript):

1) This paper shows for the first time the possibility to restore in amputees, and induce in intact subjects, the real-time texture perception of surfaces with medium-coarse geometries, using a biologically-inspired mechano-transduction approach (see previous reply). We are thus moving from contact and pressure information (presented in Raspopovic et al., 2014) to texture recognition, which is a more sophisticated kind of tactile information (Saal and Bensmaia, 2015). A detailed comparison with previous works is shown in this letter in Figure 11.

The feasibility of our method is shown during experiments involving both a transradial amputee with implanted TIME neural interface and intact subjects with electrical nerve fibers stimulation via percutaneous insertion of needle microelectrodes. Such parallel experimental study of an acute (percutaneous microstimulation) and a chronic (implanted TIME) neural stimulation technique is novel in neuroscientific and neuroprosthetic research.

2) The innovative hybrid FEM-neural model of human median nerve shows needle microstimulation and TIME stimulation similar outcomes, justifying the translation of the paradigm from acute to chronic implementation. As a consequence, microneedle experiments can be used in the future as a methodology to test new stimulation patterns and innovative artificial fingers/sensors, that afterwards are translated to amputees with implantable intraneural interfaces.

3) Together with subjective perception, the quantitative analysis of EEG activity allowed to identify the brain areas mainly targeted by the different stimuli, supporting the hypothesis that the electrical stimulation was biologically plausible (i.e., similar to mechanical stimulation of the intact fingertip).

The first and main goal listed above has been achieved by changing the experimental setup described in the previous study (Raspopovic et al., 2014), as follows:

– Improving the quality of the sensors embedded in the fingertip to gather more complex tactile patterns during the tasks. This is of course a “pre-condition” to provide the complex tactile information to the user. Several recent papers (Zang et al., 2015, Tee et al., 2015) even if extremely interesting are able to provide only one-dimensional pressure information. This paper will be the first one showing in neuroprosthetics the potentials of triaxial sensors, that can resolve the normal and shear components of mechanical interaction (with a focus on the shear component, in this study) applying a computational model to achieve neuromorphic spiking transduction. This transduction mechanism is successfully used with human subjects for the first time in this paper, whereas the feasibility only in artificial touch studies had been previously shown (as necessary from scientific and ethical viewpoints to avoid exposing the human subjects to unnecessary invasive procedures). Here, we show that the human nervous system can make use of such information as machine-learning methods could with fully robotic experiments not involving humans in the loop.

– In conclusion, as shown in the “Encoding variable” column of Figure 11, the way the stimuli encode in the pulses here is complementary to the approach followed in the previous study (Raspopovic et al., 2014). There, the delivery of current pulses to the nerve had a fixed frequency and duration within each session and was modulated in amplitude, thus temporal coding had not been implemented. In such case the number of afferent fibers recruited grew monotonically with the stimulation charge and hence the perceived force modulates accordingly, as discussed in a recent review (Saal and Bensmaia, 2015). Conversely, in the present study percepts about textural features were induced modulating the temporal structure of the spiking activity delivered through the nerve without modulating the pulses amplitude (nor the pure firing rate as shown by the analysis of Figure 10 in the manuscript).

– Overall, these are major hardware and computational differences with respect to our previous paper, and were introduced in this study in order to target different qualities of tactile perception, showing, for the first time, feasibility of restoration of texture recognition with “bionic” hand prostheses.

Author response image 1.Comparison of tactile neural sensory feedback techniques across literature.Induced tactile sensations are classified as in a recent review (Saal and Bensmaia, 2015).**DOI:**
http://dx.doi.org/10.7554/eLife.09148.022

Third, the work presents tactile encoding – in particular coarse + geometric -> shape sensing. This is not clear. Is it only the artifact of their stimulus (grating) or is this the result of perception by the subjects – having tested many stimuli. i.e. is the perceptual output of the given microneurographic access to nerves and penetrating electrodes' access to the fascicles? If so, how specific and representative is it?

The goal of the study was to verify that using intraneural stimulation subjects can gather information about textural features of medium-coarse stimuli. As discussed before, the spiking activity is generated according to the type of mechanical stimuli presented and we expect generalization capability with different textures. The following points better clarify this aspect in details and discuss the changes made to the paper:

1) All the subjects were asked to qualitatively evaluate the experienced stimulation, and all could perceive the received electrical neural stimulation as similar to mechanical alternation between ridges and gaps. Such point was better clarified in the text.

2) To better clarify how we delivered the neural stimulation patterns to the subjects, we decided to change the labels of the experimented surfaces from the previous A1, A2, B1, B2, C1, C2, D1, D2 to ΔSP+/-, where ΔSP is the difference (in mm) in spatial period between the two halves of the surface and + or – indicates whether the spatial period is increasing or decreasing (with + or – for the constant stimulus having no effect, but being used because the surface was presented half times in one side and half of occurrences in the other side to avoid polarizing the experiment in case of fabrication asymmetries). Labeling of surfaces was then changed to: Δ2.5+, Δ2.0+, Δ1.0+, Δ0.0+, Δ0.0-, Δ1.0-, Δ2.0-, Δ2.5- (as shown in the revised manuscript in Figure 1, Figure 3, Figure 9 and Figure 10).

3) Furthermore, the stimuli are now sorted with respect to such difference (with sign) between the two halves of the stimulus, so that the raster plot (Figure 1) and confusion matrices (Figure 3 and Figure 10) could now be more straightforward to read.

4) In Figure 1 of the manuscript, the raster plot represents the spikes that were generated while scanning the gratings on the sensorized region of the artificial fingertip, as represented by the mechano-neuro-transduction (MNT) process. Such spikes are delivered to the neural pathways to elicit artificial percepts by means of implanted TIME interface for the amputee subject or by means of the needle electrode for the intact subjects.

Fourth, the statistical analysis of the data should be improved. The text and Figure 1 discuss the results of the study in terms of the overall success rate. Figure 3 shows the breakdown of the responses by subject. There is a conspicuous lack of analysis of the data. The data should be analyzed not by lumping together all of the subjects and reporting an aggregate successful discrimination rate, but rather by individual subject as a function of stimulus type. The coarser the stimulus, the greater the discriminatory power and the weaker the stimulus the less the discriminatory power. There should be an analysis showing that the performance by individual is greater than what would be expected by chance, where chance here is 0.33. If there is an approximate monotone relation among the stimuli then a logistic regression analysis could be performed. The analysis should be conducted using confidence intervals or Bayesian methods and not just simply by reporting p-values. An advantage of a Bayesian hierarchical analysis or a random effects model across subjects is that they provide a formal way of pooling information across subjects. The latter is crucial for this problem because the authors which to establish that they have been able to provide neuromorphically tactile sensation. This statement should be made with an assessment of the accuracy of the discriminatory power of the MNT technique for each subject as a function of stimulus type. It is likely that the results will reveal that more trials will be required to show that there is truly discriminatory capability.

We carefully followed all such constructive comments made by the reviewers, as discussed below:

1) First, in the revised manuscript we computed a statistical analysis per individual subject, as a function of stimulus type, showing that the performance by individuals is greater than what would be expected by chance, conducted using confidence intervals. As shown in the new panels, added to Figure 3 (illustrated below), for all four subjects involved in the percutaneous electrical microstimulation protocol, we found that the 95% confidence interval for the fraction of correct answers (Clopper Pearson exact interval) was entirely above chance level for the surfaces characterized by 2.5 mm and 2.0 mm differences in spatial period ∆SP between the two halves. For subject DAS implanted with TIME interface, the 95% confidence interval was largely above chance level for each single stimulus (see the new panels, added to Figure 10, illustrated below). Significance is particularly evident in the results with the amputee subject, since the chronically-implanted interface allowed carrying out a quite high number of trials, whereas the percutaneous electrical microstimulation in intact subjects is a technique prone to interruption due to the possible sudden loss of contact between the electrode and the nerve. Hence, with acute microstimulation the number of trials could not be arbitrarily increased on a per-subject basis but still the current sets of trials provide evidence that the stimulations induced significant discriminatory capability in each subject.

2) We thank the reviewers for the constructive suggestion to perform a logistic regression analysis to investigate whether there was an approximate monotone relationship between the rate of correct response and the difference in spatial coarseness of the perceived stimuli. This analysis is shown in the new panels added to Figure 3 and Figure 10.

The authors state that they observed no difference in the natural EEG activity and evoked stimulation in source topography, response timing and clustering. How were these assessments made? They involve negative results and require a power analysis to establish their validity.

We thank the reviewers for addressing this important issue. In the revised manuscript we now clearly specify each test applied to assess statistical significance. We also ran the requested power analysis for each comparison, to investigate the validity of the results.

More specifically, with respect to the response timing, the test used for the statistical comparison was a Montecarlo statistics with cluster correction for multiple comparisons. It is now specified in the Methods (in the section entitled EEG signal processing), as follows:

“Event-related potentials (ERPs) were time-locked to the onset of either the electrical microstimulation or of the sliding phase of the mechanical stimulation. ERPs were normalized for the standard deviation of the prestimulus (1000 ms). ERP’s statistical significance between conditions (electrical microstimulation vs. mechanical stimulation) was assessed using a Montecarlo statistics with cluster correction for multiple comparisons (triangulation and maxsum as clustered statistics)(Maris and Oostenveld, 2007), adapted from the FieldTrip toolbox (Oostenveld et al., 2011).”

As suggested by the reviewers, we computed a power analysis (GPower, Duesseldorf, Germany) on the ERPs (Figure 4—figure supplement 1) depicted in Figure 4 of the manuscript, with the following comparisons:

1) Sample size computation based on the effect size of the prestimulus and the ERP evoked by electrical microstimulation;

2) Sample size computation based on the effect size of the prestimulus and the ERP evoked by mechanical stimulation;

3) Sample size computation based on the effect size of the prestimulus recorded during the sessions with electrical microstimulation and of the prestimulus recorded during the sessions with mechanical stimulation;

4) Sample size computation based on the effect size of the ERP evoked during the sessions with electrical microstimulation and of the ERP evoked during the sessions with mechanical stimulation.

For the above conditions 1 (prestimulus vs. ERP evoked during electrical microstimulation) and 2 (prestimulus vs. ERP evoked during mechanical stimulation), with our sample size of 4 subjects the power of the test was 0.75 and 0.79, respectively, as illustrated below in the Figure 4—figure supplement 2 and Figure 4—figure supplement 3.

Conversely, the sample size required to reach a comparable power was greater than 75 subjects for both the above conditions 3 (prestimulus recorded during electrical microstimulation sessions vs. prestimulus recorded during mechanical stimulation sessions) and 4 (ERP evoked during electrical microstimulation sessions vs. ERP evoked during mechanical stimulation sessions), as illustrated below in the Figure 4—figure supplement 4 and Figure 4—figure supplement 5.

We therefore conclude that despite the small sample size, our data are sufficient to statistically highlight the activity evoked by both electrical and mechanical stimulations compared to the prestimulus baselines, and that the lack of significant differences in the ERP triggered by electrical and mechanical stimulations is as valid as it is the lack of differences between the corresponding baselines.

Although the data are more compelling for the TIME technique applied to the transradial amputee patient, a similar analysis should be performed for these data as well.

Unfortunately we do not have EEG recordings for the experiments with the amputee subject, but only those from the intact volunteers (a still completely original set of data). Furthermore, now the implanted electrodes have been removed from the stump of the amputee (though being still very well performing at the end of the experiments) as planned in the protocol approved by the ethical committee, preventing the possibility of further recordings with DAS subject.

The authors wish to establish from their simulation analyses that the response patterns under MNT and TIME are similar if not effectively the same. It is unclear where the uncertainty comes from in these analyses since they are simulation models. Therefore how can statistical assessments of uncertainty and formal statistical inferences be made? Moreover, the authors wish to infer that the stimulation responses of the two modalities do not differ. They report a p-value without a specific statement about the statistical test being used in the analysis. Because the authors wish to establish a negative result of no difference between the two stimulations modalities, they should report a power analysis stating what types and magnitudes of differences their investigations were calibrated to detect.

The present model is a realistic computational model of the human median nerve, implemented starting from the realistic anatomical picture and fiber distribution data (as illustrated in Figure 6—figure supplement 1).

The uncertainty for the two simulated devices is explained in the continuation: a total amount of 100 modeled fibers for each fascicle were placed randomly in the specific target fascicle, for several placements (Figure 8 of the manuscript, Figure 12 and Figure 8—figure supplement 1 added to this letter). As the fiber organization within different fascicles in the nerve is unknown, we assumed that fibers within single fascicles belong to the same functionality. Moreover, since there is an inherent anatomical uncertainty in their placement and extension, for every position of stimulating electrode/needle we implemented nine populations having different extensions and centroids for the same fascicle (i.e., spanning from having the whole fascicle uniformly populated, to the case of very concentrated population where the fibers are almost touching each-other, Figure 8 of the manuscript, Figure 8—figure supplement 1). This is carried out since a-priori it is unknown whether the population of interest is concentrated in the small portion of a fascicle, or possibly uniformly distributed over it, and by simulating different populations placement we are emulating the inherent biological variability. Then the analysis has been performed for wide range of cases (for small, medium and big fascicles, with devices placed within, close, or far away from it), finally resulting in n=90 simulations for TIME and n=45 simulations cases for microneedle (this difference is due to the fact that in every position implemented for one needle tip the TIME electrode had 2 sites: right and left ones, at the same position).

The stimulations effects achieved by using microneedle and TIME interfaces (an example for one TIME active site is shown in this letter in Figure 12) were simulated, in order to understand whether the results obtained with microneedle in intact subjects could be translated (in terms of recruited population and respective current necessary) to TIME interface in an amputee.

Author response image 2.An example of the simulated stimulation of neural fibers by means of TIME interface, with the active site of the electrode close to, but outside from, the fascicle of interest (different populations are under-represented for the sake of proper visualization).**DOI:**
http://dx.doi.org/10.7554/eLife.09148.023

The model indicated that the stimulated portion of axonal population is similar for the two interfaces. In fact, we evaluated the statistical similarity between the injected charges necessary to reach the threshold of recruitment (defined at 10% of recruited fibers among the simulated population) for the two devices. The hypothesis of normality was studied for the groups resulting in non-parametric distributions. In order to compare non-parametric distributions we chose Kruskal-Wallis test (one-way analysis of variance by ranks). It is a non-parametric test, suitable for the case of populations with not equal number of samples, and we chose the threshold p-value of 0.05. Such information was now integrated in the revised manuscript.

Our statistical test is comparing the values of charge necessary to achieve a recruitment of 10% of fibers, and is tuned to take into account 5 nC as the magnitude of difference (that has been shown in the precedent stimulation studies to be the minimum range “differentiable”). For such a value, our sample of 135 overall simulations holds the power of test at 0.95 (G-Power, Dusseldorf, Germany).

The authors do not have to develop an ancillary classifier model to understand the performance characteristics of the MNT and the TIME paradigms. If they build statistical models to analyze the data from the subjects that include covariates that has IBI they will get their answer with the same model being used to analyze the data. This approach would obviate the current analyses which lead to multiple comparisons corrections.

We thank the reviewers for their comment that, jointly with the fourth point above, triggered us to complement the analysis of the psychophysical results and their correlation with electrophysiological data. As described in the reply to the fourth comment and figures therein we assessed the correlation between stimulus coarseness and the decoding performance for both microstimulation with intact subjects and implanted neural interface with transradial amputee. This is now shown in the revised version of Figure 3 and Figure 10, and discussed in the manuscript. In the manuscript we also report that a logistic fit of performance works for IBI but not for average firing rate since the latter does not correlate with coarseness.

[Editors' note: further revisions were requested prior to acceptance, as described below.]

Suggested Revisions:

1) The claim in the Abstract is still too broad: "Intraneural MNT-based stimulation restored discrimination of textural features, thus enhancing the user's tactile capabilities." The word "restore" is too strong. It implies permanent restoration/recovery. Here a texture perception has been mimicked, and performance has not been "enhanced."

We agree with the reviewers. Actually, the long-term restoration/recovery of tactile sensory skills for upper limb amputees is one of our next scientific and technological objectives. Hence, following the invitation to revise the wording, we changed “restore” to “elicit”. The changes were operated to the title, the Abstract and through all the manuscript when related to the specific findings of this work. Somewhere in the manuscript “restored” was left, particularly when referring to long-term objectives (e.g., “the restoration of sensory perception is the crucial step to achieve in the development of the next generation of artificial limbs and hand prostheses in particular”). The sentence “thus enhancing the user’s tactile capabilities” has been removed from the Abstract since it was an unnecessary comment.

2) Paragraph two, subheading “Analysis of neural coding strategies”: This information pertains to Figure 10. The message that the rate (in e, worse) vs temporal code (in d, good) is getting lost in this paragraph. This is an important observation and clarification would help. The figure caption could be more explanatory too.

We thank the reviewers for triggering the revision of such paragraph in the manuscript in order to point out the message that the temporal structure of the injected spike trains can explain behavioural results, whereas a rate code could not. The revised text is reported hereafter.

“The abovementioned results show that DAS behavioral responses may be based on the temporal structure of stimulation. […]This further strengthens the hypothesis that the subject exploited the temporal structure of the response to discriminate the stimuli.”

Furthermore, the caption of Figure 10 was unified to induce a combined read of the two panels. We also modified the title of the two panels to made their content clearer.

*3) Paragraph two, subheading “Hybrid electrical-biophysical model of the median nerve for the comparison between microstimulation needle and implanted TIME”*: *As the fiber organization within different fascicles in the nerve is unknown, we assumed that fibers within the fascicles belong to same functionality.*

Please clarify how nociceptive fibers were separated from sensory (e.g. see the comment pertaining to the model). Indeed, this issue or not being able to separate different fiber types is quite critical and must be marked as a limitation.

We thank the reviewers for this very useful observation that triggered us to change the wording within the text accordingly. Indeed when stating that the fibers within the same fascicle belong to same functionality, we meant that they are innervating the same area (e.g. the tip of second finger), not that they are of the same type. Now this is corrected. While as regarding the nociceptive fibers, they are the ones with the smallest diameter and are therefore the last to be stimulated, as shown in previous studies (McNeal, 1976). Since in our study and model we were more interested about the recruitment at the threshold or at half-range of sensation, the nociceptive fibers stimulation is not induced. However, as suggested by reviewers, in the revised text we also introduced into the list of the limitations (called model assumptions), the one regarding the separation of the different fiber types. The related revised text is reported hereafter.

“A list of plausible assumptions had to be taken, during the model construction. […]One of the assumptions constraining the model was that the stimulation of different fiber types, such as nociceptive fibers, was not induced, at used current range.”

4) Figure 2: Traces and colors are not well explained.

The caption of Figure 2 was revised.

5) Figure 6 caption: The X marker represents the targeted fascicle where the fiber activation per 9 different populations was calculated. What does this mean (what populations?) for different locations of microneedle and TIME? This procedure is carried out analogously for medium and small fascicles, confirming results. Please could you also clarify the procedure.

This point was now better explained within the revised manuscript, in the Materials and methods section, in the caption of Figure 6, of Figure 8 and of Figure 8—figure supplement 1. Since there is an inherent uncertainty in the placement and extension of fibers innervating a specific hand district, for every position of stimulating electrode/needle we implemented 9 populations having different extensions and centroids for the same large fascicle (i.e., spanning from having the whole fascicle uniformly populated, to the case of very concentrated population where the fibers are almost touching each-other, Figure 8). Then the analysis has been performed for the range of significant fascicle sizes, defined as median size representatives of three groups: small (1 population implemented), medium (5 populations implemented) and large (9 populations). Different placements of the electrodes were simulated as well as illustrated in the figure. The captions of all figures related to this topic were changed accordingly.

6) Figure 8: outcomes. Color-coding is not explained.

Different colored dots (green, red, blue, black,…) indicate the positioning of different populations of the fibers implemented, which is emulating the unknown, possible dispersions of the positioning of fibers. This is now added to the caption of Figure 8. Figure 8—figure supplement 1 further clarifies the color-coding.

7) Figure 10: This figure, slightly modified to clarify rate vs temporal, would be much clearer (and certainly this result is important).

We thank the reviewers for pointing out this issue. The caption of Figure 10 and the related text in the manuscript were revised as discussed in the reply to point 2 above.